# Estrogen-related receptor gene expression associates with sex differences in cortical atrophy in isolated REM sleep behavior disorder

Marie Filiatrault [1,2], Violette Ayral[1,2], Christina Tremblay [1], Celine Haddad [1,3], Véronique Daneault[1], Alexandre Pastor-Bernier[1], Jean-François Gagnon[1,4,5], Ronald B. Postuma[1,6,7], Petr Dušek[8], Stanislav Mareček [8], Zsoka Varga [8], Johannes C. Klein [9], Michele T. Hu[9], Isabelle Arnulf [10], Pauline Dodet [10], Marie Vidailhet[10], Jean-Christophe Corvol [10], Stéphane Lehéricy[10], ICEBERG Study Group*, Simon Lewis[11], Elie Matar[12], Kaylena A. Ehgoetz Martens[12,13], Lachlan Churchill[12], Per Borghammer [14], Karoline Knudsen[14], Allan K. Hansen[14], Dario Arnaldi[15,16], Beatrice Orso[15,16], Pietro Mattioli[15,16], Luca Roccatagliata[16,17] & Shady Rahayel [1,18] ✉

Isolated REM sleep behavior disorder, characterized by dream-enacting movements during REM sleep, is a male-predominant parasomnia and the strongest prodromal marker of synucleinopathies. Individuals with this disorder show cortical atrophy whose regional distribution covaries with gene expression patterns measured in the healthy human brain. However, the effect of sex on these brain changes remains unknown. The study objective is to comprehensively assess sex differences in cortical morphology and to characterize the healthy-brain gene expression correlates of brain abnormalities using the largest international multicentric MRI dataset of polysomnography-confirmed patients. Males have significantly more extensive and severe cortical thinning compared to females, despite similar age and clinical features. Imaging transcriptomics analyses indicate that regions affected in female patients map onto areas with higher expression of estrogen-related receptor genes, particularly *ESRRG* and *ESRRA*, in the healthy brain. These findings support potential sex-specific neuroprotection in the prodromal stages of synucleinopathies and may inform personalized and targeted therapeutic strategies.

Isolated REM sleep behavior disorder (iRBD) is a parasomnia characterized by the loss of REM sleep muscle atonia, leading to dream-enacting behaviors such as vocalizations, limb movements, and complex, often violent, movements that closely align with dream content[1,2].

Critically, iRBD is the strongest known prodromal marker of neurodegenerative synucleinopathies, with over 70% of individuals eventually developing dementia with Lewy bodies (DLB), Parkinson's disease (PD), or in a smaller proportion, multiple system atrophy

A full list of affiliations appears at the end of the paper. *A list of authors and their affiliations appears at the end of the paper.
✉e-mail: shady.rahayel@umontreal.ca

(MSA)[3–5]. Phenoconversion can take up to 15 years[3,4], offering an important window for studying early biomarkers and disease mechanisms before overt neurodegeneration emerges[6].

Neuroimaging studies show that individuals with iRBD already exhibit cortical atrophy compared to healthy controls[7,8], which is associated with cognitive decline and motor impairments[9,10]. While definite phenoconversion biomarkers are still being investigated, a specific atrophy signature can predict whether an individual with iRBD is more likely to convert to DLB rather than PD[11]. In iRBD, brain atrophy, such as reduction in cortical thickness, follows distinct patterns, potentially driven by prion-like propagation along neural networks and selective regional vulnerability[8,12]. To identify potential selective vulnerability factors in relation to neurodegenerative changes in specific brain regions, imaging transcriptomics has emerged as a powerful approach in various neurodegenerative conditions[13]. This method integrates regional neuroimaging measures with spatial gene expression profiles from post-mortem human brain atlases to uncover transcriptomic signatures associated with regional brain changes[14]. In iRBD and PD, MRI-derived atrophy occurs in regions overexpressing genes linked to mitochondrial functions and macroautophagy, two processes strongly affected in these disorders[15,16]. In PD, regional accumulation of iron measured through quantitative susceptibility mapping was related to higher regional expression of genes associated with metal detoxification and synaptic function[17]. In MSA, regions with atrophy on MRI showed genetic overexpression of oligodendrocytes, the cells that harbor the glial cytoplasmic inclusions characteristic of this disorder[18]. In Alzheimer's disease, regional atrophy occurred in regions enriched for genes associated with protein remodeling processes, with *APOE*, a gene strongly associated with Alzheimer's disease, ranking among the most associated genes [16]. Furthermore, a recent study showed that the brain regions most vulnerable in terms of MRI atrophy to declining kidney function overexpressed angiotensinogen-related genes[19], providing a potential linkage for the brain–kidney axis. Taken together, this supports the ability of imaging transcriptomics to pinpoint some of the vulnerability mechanisms overexpressed in regions undergoing pathological changes in neurodegenerative diseases.

Sex is another key factor influencing neurodegeneration[20], yet its role in brain changes associated with iRBD remains largely unexplored. iRBD has a strong male predominance, with reported prevalence ratios as high as 8:1[4,21], leading to an underrepresentation of female individuals in iRBD studies, particularly in neuroimaging studies. In manifest synucleinopathies, sex differences are well documented[22]. In DLB and PD, studies highlight that males show earlier onset[23,24], greater motor impairment[24,25], and more extensive and severe neurodegenerative brain changes[26,27], while females exhibit relative neuroprotection, possibly mediated by estrogens[20]. Estrogens and estrogen-related functions play a key role in mitochondrial function and dopaminergic neuron survival[28], potentially delaying neurodegeneration. These protective effects, supported by human and animal studies[29,30], have been suggested as a possible explanation for the milder cortical atrophy, slower and more benign disease progression in females compared to males with DLB and PD[20,31]. However, whether such sex-related neuroprotective mechanisms already operate in early synucleinopathies remains unknown.

In iRBD, studies on sex effects are scarce and yield conflicting results[32]. While some studies suggest that females have a later onset age, others report earlier onset or no difference[32]. Some behavioral studies indicate that females exhibit less aggressive dream-enacting behaviors and fewer sleep-related injuries compared to males[33], although findings are inconsistent[22]. Sleep architecture differences, such as shorter N1 sleep and longer REM latency in females, have also been noted[34]. Beyond these observations, large longitudinal studies have, however, failed to detect sex-related differences in disease progression[4,5], leaving a gap in understanding how sex may influence disease trajectory. Studies using neuroimaging have demonstrated a complex interplay between sex and brain changes in iRBD. Our group previously identified signatures of brain atrophy and perfusion predictive of phenoconversion in iRBD[11,35]. While the atrophy pattern predictive of DLB in iRBD was not influenced by sex, the pattern of deformation primarily localized in the brainstem and associated with REM sleep motor activity was more associated with being male[11]. The perfusion signature was also more pronounced in males[35]. These findings highlight the need for further research into sex-specific neurodegenerative patterns in iRBD.

In this study, we used a large international, multicenter dataset of 888 brain MRI scans (408 polysomnography-confirmed iRBD patients and 480 healthy controls) to investigate sex-related differences in atrophy. Using vertex-based cortical surface analysis, we investigated whether sex interacted differently with cortical thickness between iRBD patients and controls, on a total of 687 participants passing eligibility and quality control criteria. We quantified the extent of cortical thinning in male and female iRBD patients matched for age and clinical features and used imaging transcriptomics to investigate whether gene expression patterns measured in the healthy adult brain aligned with sex-related atrophy differences. Gene enrichment analysis was further performed to identify molecular pathways overrepresented in regions showing sex effects. To investigate the systemic relevance of implicated genes, we examined their normative expression across peripheral and central tissues in the Genotype-Tissue Expression (GTEx) Project. We hypothesized that males with iRBD would exhibit greater cortical thinning than females, regions relatively less affected in females would coincide with higher expression in the healthy brain of genes involved in estrogen-related molecular functions.

## Results

### Demographics and clinical characteristics

Of the 888 eligible participants (408 iRBD patients and 480 controls), 7 (0.8%) did not pass processing (four iRBD patients, three controls) and 134 (15.1%) did not pass quality control (61 iRBD patients, 73 controls) based on published criteria (Fig. 1 for a flowchart)[36]. Of the 65 iRBD

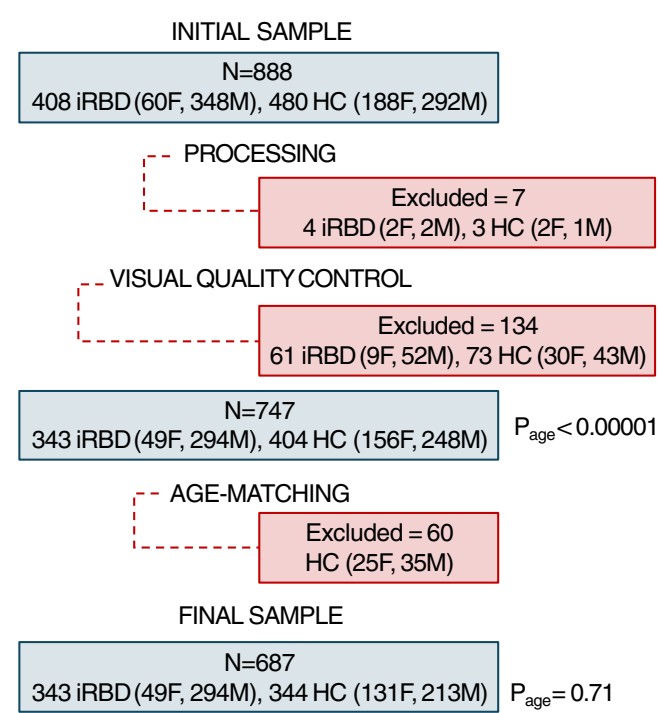

**Fig. 1 | Participant selection flowchart.** The initial sample included 888 participants. Following exclusions during MRI processing, visual quality control, and age matching, the final sample consisted of 687 participants. *P* values (two-sided) indicate differences in age distributions between groups before and after matching using independent two-sample *t* tests. F = female; HC = healthy controls; iRBD = isolated REM sleep behavior disorder; M = male.

patients, 11 were females and 54 were males. The excluded iRBD patients did not differ significantly in age ($P = 0.31$) and sex proportion ($P = 0.91$) from those who passed these processing steps. However, the resulting iRBD group significantly differed in age compared to controls (iRBD patients: $67.0 \pm 6.9$, controls: $63.7 \pm 9.6$, t[726] = 5.38, $P < 0.00001$, $d = 0.39$, mean difference [95% CI] = 3.26 [2.04, 4.48]). Groups were therefore matched for age, which led to the exclusion of 60 controls (<54 years old), namely 25 females and 35 males, resulting in an age-matched final sample of 343 iRBD and 344 controls for analysis (iRBD patients: $67.0 \pm 6.9$; controls: $66.6 \pm 6.9$, $P = 0.71$). The iRBD group comprised 49 (14%) females and 294 (86%) males, while the control group included 131 (38%) females and 213 (62%) males. As expected[21], sex distribution significantly differed between iRBD patients and controls ($\chi^2[1]$ = 50.3, $P < 0.0001$). However, this difference did not impact analyzes, as all comparisons were either stratified by sex or performed on W-scored measurements, which use controls as the reference group.

In terms of clinical variables, iRBD patients had higher MDS-UPDRS-III scores ($U = 17826$, $Z = -9.72$, $P < 0.001$) and lower MoCA scores ($U = 30251$, $Z = -5.81$, $P < 0.001$) compared to control participants (Table 1). Within the iRBD group, no significant differences between females and males were observed for age ($P = 0.39$), MDS-UPDRS-III scores ($5.5 \pm 5.2$ in females vs. $6.7 \pm 6.0$ in males, $P = 0.17$) or MoCA scores ($25.6 \pm 3.9$ in females vs. $25.5 \pm 3.0$ in males, $P = 0.18$, range: 13–30) (Table 1). No differences were also found for age of onset of RBD symptoms ($P = 0.45$), self-reported duration of RBD symptoms ($P = 0.34$), age at video-polysomnography-confirmed diagnosis of RBD ($P = 0.46$), RBD duration since diagnosis ($P = 0.50$), and proportion of patients taking RBD medication (clonazepam or melatonin) ($P = 0.45$). In terms of performance on olfactory identification tasks, there were no significant differences between iRBD males and females on each assessment scale and on the scores converted into the 16-item Sniffin' Sticks[37], although iRBD females ($9.4 \pm 3.7$, score on 16) tended to have better performance than iRBD males ($8.5 \pm 3.5$, t[273] = −1.38, $P = 0.085$, d = 0.24, mean difference [95% CI] = −0.85 [−2.05, 0.36]). In contrast, iRBD females had significantly less years of education compared to iRBD males ($12.7 \pm 3.6$ in females vs. $14.5 \pm 3.5$ in males, t[319] = 3.18, $P = 0.002$, $d = 0.52$, mean difference [95% CI] = 1.83 [0.69, 2.96]). Within the control group, there were no significant differences between females and males in age ($P = 0.16$) and MDS-UPDRS-III ($P = 0.93$), but males had significantly lower MoCA scores compared to females ($27.4 \pm 2.5$ in females vs. $26.8 \pm 2.2$ in males, $U = 5744$, $Z = -3.18$, $P = 0.001$, range: 19–30) (Table 1). Using a MoCA threshold of ≤25 to define possible mild cognitive impairment[38], the proportion of possible mild cognitive impairment was significantly higher in the iRBD group (42%; 33% of iRBD females and 44% of iRBD males) compared to the control group (22%; 17% females, 26% males) ($\chi^2[1]$ = 30.30, $P < 0.001$). This proportion did not differ based on sex in either the iRBD or the control groups (iRBD: $P = 0.20$, controls: $P = 0.15$).

## Sex impacts cortical thickness differently in iRBD and controls

Before examining sex effects on cortical atrophy (i.e., reductions in cortical thickness beyond what is expected for age) in iRBD, we first assessed whether sex differences in cortical thickness varied between iRBD patients and controls. Vertex-wise surface analysis of cortical thickness revealed a significant sex-by-group interaction in three cortical clusters (Fig. 2A, Table 2). The peaks were located in the left posterior cingulate cortex ($7846\,\text{mm}^2$, 16820 vertices, Talairach coordinates: $x = -4.7$, $y = -27.9$, $z = 34.6$, $P = 0.0007$) and superior parietal cortex ($3279\,\text{mm}^2$, 8099 vertices, $x = -32.0$, $y = -38.9$, $z = 39.2$, $P = 0.0004$), extending to sensorimotor and medial frontal cortex, and in the right paracentral cortex ($8607\,\text{mm}^2$, 18097 vertices, x = 5.2, $y = -25.1$, $z = 66.3$, $P = 0.0004$), extending to the motor cortex, medial frontal cortex, and dorsolateral prefrontal cortex (Fig. 2A, Table 2). In the left posterior cingulate and right paracentral clusters, males with

iRBD showed significantly reduced cortical thickness compared to females, while no significant sex differences were observed in controls (Fig. 2B, Table S1). In contrast, in the left superior parietal cluster, males exhibited reduced cortical thickness compared to females in both groups, with this reduction being significantly more pronounced in iRBD patients (Fig. 2B). Vertex-based analyses conducted separately in iRBD patients and controls revealed similar clusters in iRBD, identifying six clusters (two in the left hemisphere, four in the right hemisphere) where males had significantly lower cortical thickness than females (Fig. S1, Table S2). In contrast, no significant male-female difference was detected in the control group. When using more stringent cluster-forming thresholds to identify significant clusters ($P < 0.01$), we still identified a sex-by-group interaction effect in participants (Fig. S2).

To verify the sex effect on other structural measures, vertex-based cortical surface area and volume analyses were conducted. Results revealed a significant sex-group interaction on cortical surface area and cortical volume in the right superior frontal cortex (Fig. S3, Table S3). When conducting analyses separately in iRBD and controls, iRBD males showed reduced cortical surface area and volume in the right superior frontal cortex compared to females (Fig. S3, Table S3). This effect was not present in controls. We also investigated the presence of a sex-by-group interaction on subcortical volumes and found no significant effect ($P_{FDR} > 0.05$) (Table S4). Taken together, these findings support a significant interaction between sex and cortical morphology in iRBD, with males showing greater thinning than females, despite similar age and clinical severity.

## Cortical atrophy differs based on sex in iRBD

We next examined whether the pattern and extent of cortical atrophy differed between females and males with iRBD using a parcel-wise approach. To address this, we parcellated the cortical surface into 1000 cortical regions across hemispheres, extracted cortical thickness values, harmonized them across acquisition sites, and standardized them for age and sex using regression models from healthy controls (W-scores). Cortical atrophy in iRBD patients was assessed using one-sample t-tests, determining whether regional values significantly deviated from 0, corresponding to expected value in age- and sex-matched controls (i.e., no atrophy compared to controls). Significant negative deviations indicated cortical thinning in iRBD patients. In the left hemisphere, iRBD males exhibited significant atrophy ($P < 0.05$) in 191 regions (38%), predominantly affecting the sensorimotor cortex, perisylvian region, and occipital cortex (Fig. 3A). Among these, 183 regions (37%) remained significant after FDR correction, with t-scores ranging from −2.3 to −12.1. In contrast, iRBD females exhibited atrophy in 53 regions (11%) regions, largely overlapping with affected regions in males, particularly in the sensorimotor and occipital cortices. However, only 4 regions (1%) remained significant after FDR correction, with t-scores ranging from −3.7 to −4.8 (Fig. 3B). In the right hemisphere, iRBD males exhibited atrophy in 298 regions (60%) ($P < 0.05$), primarily affecting the sensorimotor cortex, perisylvian region, and occipitoparietal cortex. After FDR correction, 262 regions (53%) remained significant, with t-scores ranging between −2.2 and −6.7. In iRBD females, 84 regions (17%) showed significant thinning ($P < 0.05$), again in similar areas affected in males; however, none remained significant after FDR correction. Overall, the proportion of significantly atrophied regions differed between sexes ($\chi^2[1]$ = 210.65, $P < 0.0001$), These findings reveal that cortical atrophy in iRBD is markedly less widespread and severe in females compared to males, despite comparable age and clinical severity.

## Estrogen-related receptor genes are associated with sex effects in iRBD

To investigate the mechanisms potentially protecting iRBD females from cortical atrophy, we examined whether brain regions

**Table 1 | Demographics and clinical characteristics of participants**

| Variables | iRBD (n = 343) | | | Controls (n = 344) | | | P-value | | |
|---|---|---|---|---|---|---|---|---|---|
| | Total | Females | Males | Total | Females | Males | Total[a] | iRBD[b] | Controls[c] |
| Sex, n (%) | 343 | 49 (14%) | 294 (86%) | 344 | 131 (38%) | 213 (62%) | $1.02 \times 10^{-7}$[d] | - | - |
| Age, years | 67.0 ± 6.9 | 66.3 ± 5.9 | 67.2 ± 7.1 | 66.6 ± 6.9 | 66.3 ± 5.9 | 67.0 ± 6.9 | 0.71[e] | 0.39[e] | 0.16[e] |
| Education, years | 14.2 ± 3.6 | 12.7 ± 3.6 | 14.5 ± 3.5 | - | - | - | - | 0.002[e] | - |
| Education, ≥12, yes/no | 66%/34% | 42%/57% | 54%/46% | - | - | - | - | 0.19[d] | - |
| MDS-UPDRS-III | 6.5 ± 5.9 | 5.5 ± 5.2 | 6.7 ± 6.0 | 3.1 ± 5.8 | 4.0 ± 8.3 | 2.6 ± 3.7 | <0.001[f] | 0.17[f] | 0.93[f] |
| MoCA | 25.5 ± 3.1 | 25.6 ± 3.9 | 25.5 ± 3.0 | 27.0 ± 2.3 | 27.4 ± 2.5 | 26.8 ± 2.2 | <0.001[f] | 0.18[f] | 0.001[f] |
| Possible MCI[g] | (42%) 22.8 ± 2.5 | (33%) 21.3 ± 3.4 | (44%) 23.0 ± 2.3 | (22%) 23.9 ± 1.5 | (17%) 23.8 ± 1.5 | (26%) 24 ± 1.5 | $6.51 \times 10^{-7}$[d] | 0.20[d] | 0.15[d] |
| Age of onset of RBD symptoms | 59.0 ± 9.1 | 59.2 ± 6.7 | 59.0 ± 9.5 | - | - | - | - | 0.45[e] | - |
| Self-reported duration of RBD symptoms at MRI | 7.9 ± 7.1 | 6.9 ± 4.6 | 8.1 ± 7.4 | - | - | - | - | 0.34[e] | - |
| Age at RBD diagnosis (PSG) | 65.4 ± 7.1 | 64.8 ± 6.7 | 65.6 ± 7.2 | - | - | - | - | 0.46[e] | - |
| RBD duration since diagnosis at MRI | 1.7 ± 2.2 | 1.5 ± 1.5 | 1.7 ± 2.3 | - | - | - | - | 0.50[e] | - |
| Current RBD medication (yes/no)[h] | 40% / 36% | 39% / 31% | 40% / 37% | - | - | - | - | 0.45[d] | - |
| Olfactory score | | | | | | | | | |
| Calibrated scores[i] | 8.6 ± 3.5 | 9.4 ± 3.7 | 8.5 ± 3.5 | - | - | - | - | 0.09[i] | - |
| Sniffin' Sticks, 12 items (5% dataset)[k] | 6.8 ± 3.0 | 9.0 ± 2.6 | 6.6 ± 3.0 | - | - | - | - | 0.12[i] | - |
| Sniffin' Sticks, 16 items (32% dataset)[k] | 7.6 ± 3.2 | 9.0 ± 3.7 | 7.5 ± 3.1 | - | - | - | - | 0.09[i] | - |
| UPSIT-40 (35% dataset)[k] | 22.0 ± 7.1 | 23.3 ± 6.2 | 21.7 ± 7.2 | - | - | - | - | 0.25[i] | - |
| UPSIT-12 (28% dataset)[k] | 7.2 ± 2.9 | 7.2 ± 2.8 | 7.2 ± 2.9 | - | - | - | - | 0.48[i] | - |

Data are presented as mean ±SD.
[a]iRBD patients versus controls.
[b]Female vs. male iRBD patients.
[c]Female vs. male controls.
[d]Chi-squared test.
[e]Two-sided independent two-sample t-test.
[f]Mann-Whitney U test.
[g]Using a threshold of MoCA ≤ 25 for possible MCI; the value represents the MoCA value in participants showing possible MCI.
[h]Patients taking RBD medication (clonazepam or melatonin) or no medication.
[i]Sniffin' Sticks, 12 items, UPSIT-40, and UPSIT-12 scores were converted to Sniffin's Sticks, 16 items, according to published guidelines[37].
[j]One-sided independent two-sample t-test.
[k]Percentage of the total dataset with available data.
iRBD = isolated REM sleep behavior disorder; MCI = mild cognitive impairment; MDS = Movement Disorder Society; MoCA = Montreal Cognitive Assessment; PSG = polysomnography; SD = standard deviation; UPDRS-III = Unified Parkinson's Disease Rating Scale part III; UPSIT = University of Pennsylvania Smell Identification Test

## a | vertex-wise sex-group interaction

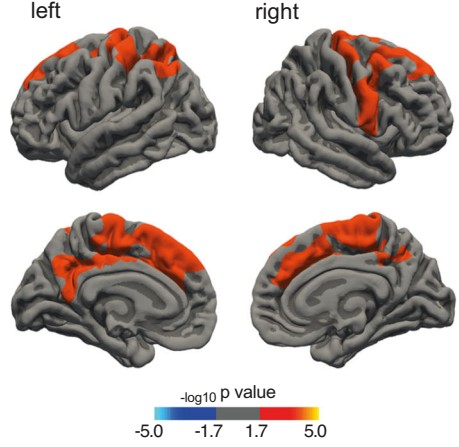

-log10 p value

-5.0   -1.7   1.7   5.0

## b | average thickness in clusters

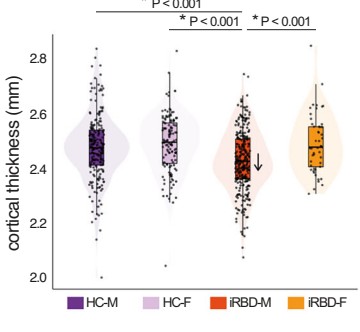

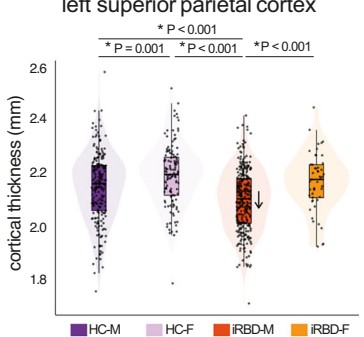

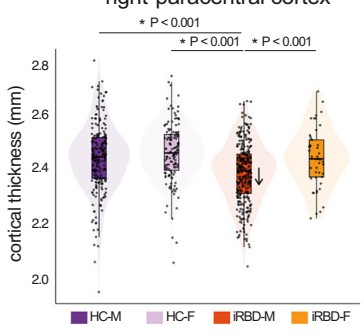

**Fig. 2 | Vertex-wise sex-by-group interaction effect on cortical thickness.**
**a** Clusters showing significant sex-by-group interaction on cortical thickness at a cluster-forming threshold of $P < 0.05$ and following Monte Carlo permutations. The color bar indicates the statistical significance on a logarithmic scale of $P$-values (−log10), with positive values (red-yellow scale) indicating the presence of a significantly stronger sex effect in iRBD patients compared to controls. **b** Average cortical thickness (in mm) across groups in significant clusters, showing significant reduction in cortical thickness in iRBD males ($N = 294$) compared to iRBD females

($N = 49$) and controls (131 females, 213 males). The $P$-values indicate significant differences between groups after one-sided independent two-sample $t$ tests. Box plots display the median (center line), the interquartile range (box; 25th-75th percentiles), and whiskers extending to the most extreme values within 1.5 times the interquartile range from each quartile. Each point represents the mean cortical thickness at cluster peak for each participant. Brain maps were generated using freeview in FreeSurfer. F = females; HC = healthy controls; iRBD = isolated REM sleep behavior disorder; M = males.

## Table 2 | Vertex-based cortical thickness analyses of sex effects

| Cluster peak location[a] | Cluster size, mm² | Number of vertices | Talairach coordinates | | | −log₁₀ $P$-value |
|---|---|---|---|---|---|---|
| | | | $x$ | $y$ | $z$ | |
| **Sex-by-group interaction** | | | | | | |
| Left posterior cingulate cortex | 7846 | 16,820 | −4.7 | −27.9 | 34.6 | 3.18 |
| Left superior parietal cortex | 3279 | 8099 | −32.0 | −38.9 | 39.2 | 3.44 |
| Right paracentral cortex | 8607 | 18,097 | 5.2 | −25.1 | 66.3 | 3.40 |
| **Sex effect in iRBD** | | | | | | |
| Left postcentral cortex | 13,676 | 29,086 | −34.5 | −32.9 | 41.1 | 5.53 |
| Left inferior parietal cortex | 2616 | 4909 | −44.6 | −73.2 | 14.4 | 4.26 |
| Right caudal middle frontal cortex | 3415 | 6690 | 37.6 | 2.3 | 34.7 | 3.89 |
| Right inferior parietal cortex | 3276 | 7191 | 41.6 | −70.3 | 36.0 | 3.16 |
| Right rostral middle frontal cortex | 2778 | 4435 | 22.2 | 52.1 | 24.1 | 2.81 |
| Right paracentral cortex | 2448 | 5660 | 7.8 | −11.9 | 60.6 | 4.09 |

Clusters were considered significant after Monte-Carlo simulation, with cluster- and vertex-level $P$-values set at $P < 0.05$.
[a]Only the region of the peak vertex is indicated (see Fig. 2 and Fig. S1 for mapping).
iRBD = isolated REM sleep behavior disorder.

with less thinning in females expressed distinct gene expression patterns that may confer neuroprotection. We first tested whether the spatial distribution of gene expression was associated with sex differences on cortical atrophy in iRBD. Using W-scored cortical thickness values, we applied a linear regression model to calculate a sex interaction estimate for each parcellated region in the left hemisphere, as gene expression data was primarily available for

this hemisphere. A positive sex interaction coefficient indicated regions where iRBD females exhibited less cortical thinning than males.

PLS regression identified two significant latent variables (LV1 and LV3) that explained significantly more covariance between gene expression and sex interaction estimates than spatially constrained null models. LV1 explained 19% of the covariance (compared to 11% in

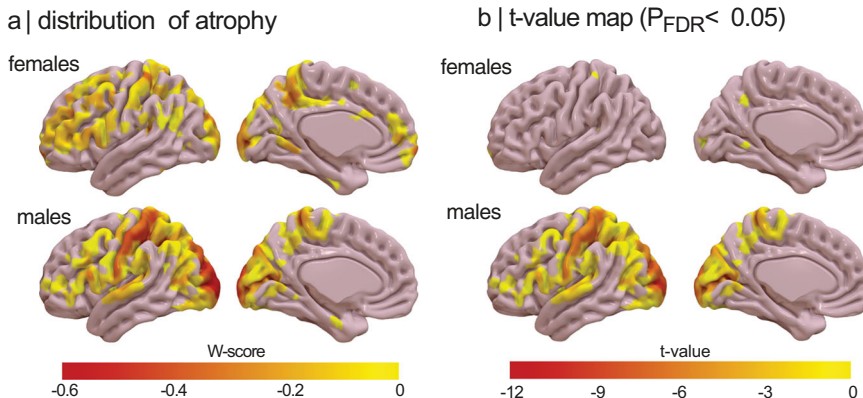

**Fig. 3 | Brain map showing sex-stratified atrophy patterns in iRBD. a** Cortical maps showing the distribution of reduced cortical thickness in females and males with iRBD against age- and sex-adjusted values. The color bar indicates W-scores, representing the severity of regional deviation compared to expected morphology. **b** Cortical maps showing the significantly atrophied regions in iRBD females and

males after conducting one sample $t$ tests for each cortical region. The color bar indicates t-values for regions reaching significance after FDR correction ($P_{FDR} < 0.05$). Cortical surface maps were generated with Surflce[77]. FDR = false discovery rate; iRBD = isolated REM sleep behavior disorder.

null models, $P = 0.007$) and LV3 explained 11% (compared to 6% in null models, $P = 0.02$) (Fig. 4A). The weights of brain regions on these significant latent variables were positively correlated with sex interaction estimates on cortical thickness (LV1: r = 0.44, $P < 0.0001$; LV3: r = 0.33, $P < 0.0001$) (Fig. 4B, C, D, F), indicating that genes positively weighted on these latent variables were overexpressed in regions where iRBD females had less cortical thinning than males.

To identify molecular functions associated with less cortical atrophy in iRBD females, we used bootstrapping to estimate the robustness of each gene's contribution to LV1 and LV3. For LV1, gene set enrichment analysis revealed significant enrichment in nuclear steroid receptor activity (8/20 genes, 40%, normalized enrichment score [NES] = 1.95, $P_{FDR} = 0.027$), which was associated with less cortical thinning in iRBD females (Fig. 4E, Table 3). The most strongly associated genes in this function were *ESRRG* (bootstrap ratio = 13.33) and *ESRRA* (bootstrap ratio = 10.92), encoding for estrogen-related receptors gamma and alpha, followed by various nuclear receptor subfamily group members (*NR3C2, NR2C1, NR3C1*) and *PPARD* (bootstrap ratio = 7.26), encoding for peroxisome proliferator-activated receptor delta (Table 4). Regional gene expression patterns for the whole group of donors, and for male and female donors separately are shown in Fig. S4. For LV3, gene set enrichment analysis also uncovered a molecular function significantly associated with positive sex interaction estimates, corresponding to reduced cortical thinning in iRBD females (Fig. 4G). The most strongly associated gene terms were enriched in olfactory receptor activity (29/48 genes, 60%, NES = 2.06, $P_{FDR} = 0.015$) (Fig. 4G, Tables 5 and S5). No significant biological process or cellular component terms were associated with LV1 or LV3.

To verify that these findings were not driven by gene ontology platform selection, we conducted a secondary enrichment analysis using GOrilla[39]. For LV1, steroid hormone receptor activity ($P_{FDR} = 0.005$, enrichment score = 14.34) and nuclear steroid receptor activity ($P_{FDR} = 0.01$, enrichment score = 6.14) emerged as the top enriched functions (Table S6), both including *ESRRG* and *ESRRA* (Tables S7 and S8). No significant biological process or cellular component terms were identified for LV1 or LV3. Taken together, these results suggest that regions where iRBD females show less cortical thinning overexpress genes enriched in nuclear steroid receptors, particularly estrogen-related receptors (*ESRRG* and *ESRRA*).

### Brain-enriched expression of *ESRRG* in sex-related cortical atrophy
To further explore the systemic roles and biological relevance of *ESRRG* and *ESRRA* in sex-related cortical atrophy in iRBD, we analyzed

their tissue-specific expression patterns using GTEx data. The GTEx dataset provides gene expression profiles from 54 non-diseased tissue types, collected from nearly 1000 post-mortem donors[40], enabling a comprehensive investigation of how these genes are expressed across human tissues.

The results revealed distinct expression profiles for *ESRRG* and *ESRRA* (Fig. 5). Violin plots showed qualitative differences: *ESRRG* exhibited a more brain-enriched expression pattern, whereas *ESRRA* showed a more ubiquitous distribution, with lower expression in the brain compared to other organs. Quantitative analysis confirmed these observations. Among tissues where each gene was overexpressed relative to the median expression across all tissues, *ESRRG* was significantly more enriched in brain tissues compared to *ESRRA* (13/27 (48%) tissue types for *ESRRG* vs. 2/27 (7%) for *ESRRA*, $\chi^2[1] = 9.23$, $P = 0.002$). *PPARD* also showed a ubiquitous distribution across tissue types (Fig. S5). Taken together, these findings highlight the brain relevance of *ESRRG*, suggesting its potential role in the selective protection of brain regions to sex-related cortical atrophy in iRBD.

### Discussion
This study investigated sex-related differences in cortical atrophy in iRBD, a prodromal stage of synucleinopathies, using a large multi-center MRI dataset of polysomnography-confirmed iRBD patients and controls. Our findings demonstrate that females with iRBD exhibit significantly less cortical thinning than males, despite similar age and clinical severity, with this effect absent in healthy controls. Although the spatial pattern of atrophy was largely similar between iRBD females and males, females exhibited more restricted and less severe atrophy, suggesting the presence of sex-specific protective mechanisms that may mitigate brain neurodegeneration in females. Transcriptomic analyses revealed that regions showing less cortical thinning in females, notably in occipital and sensorimotor areas, overlapped with areas enriched for nuclear steroid receptor functions, with *ESRRG* displaying a more brain-specific expression pattern. These findings suggest that selective protection mechanisms, potentially mediated by estrogen-related pathways, contribute to reduced neurodegeneration in iRBD females compared to males.

Sex differences in manifest synucleinopathies have been widely reported[22], yet their presence in prodromal stages such as iRBD has not been well characterized. Here, we demonstrate that males with iRBD exhibit significantly greater cortical thinning than females, despite comparable age and clinical severity. Importantly, this sex difference was absent in control participants. Our findings align with studies in manifest DLB and PD, which report greater neurodegeneration in

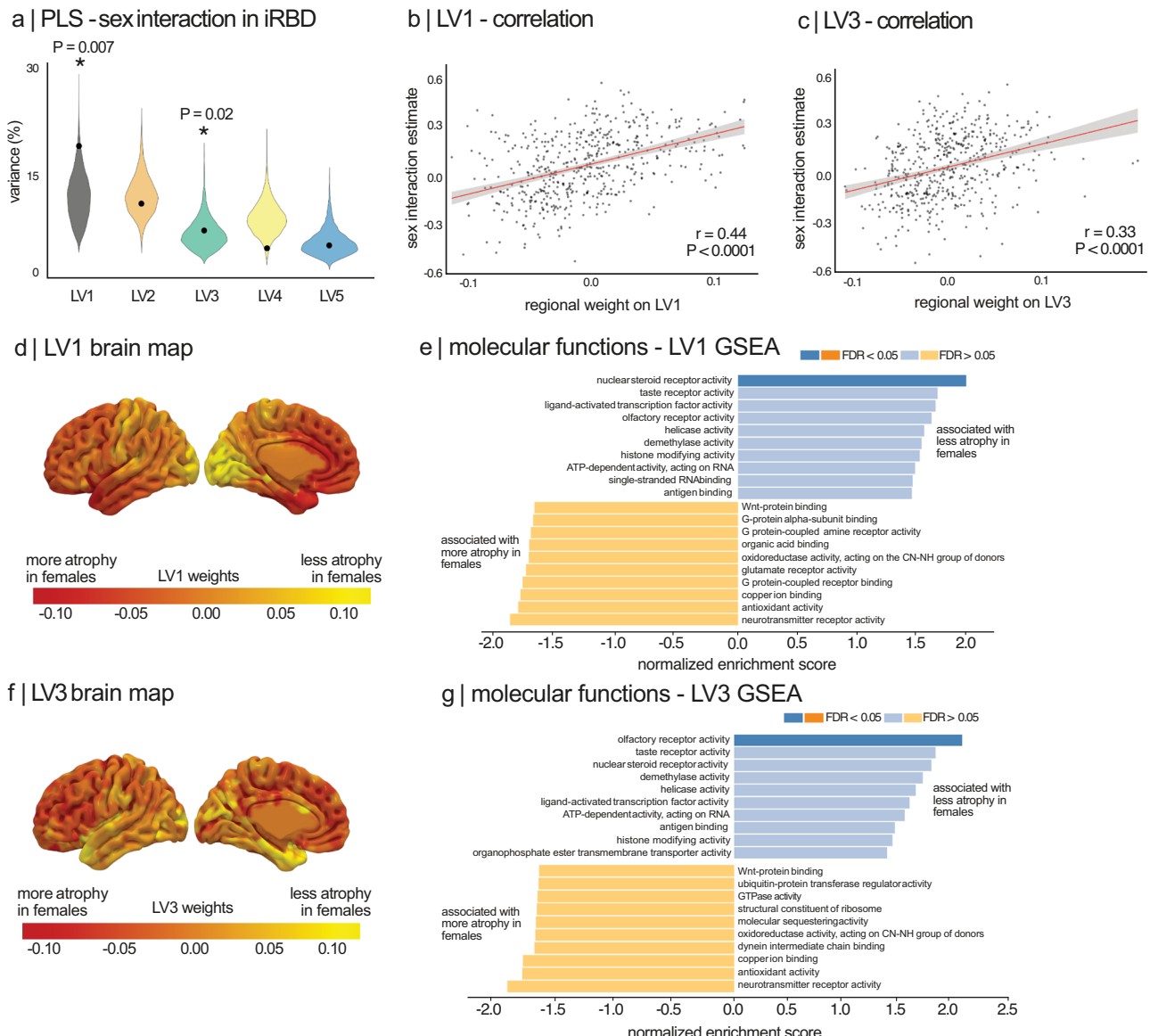

**Fig. 4 | GSEA on sex interaction estimates of cortical thickness in iRBD. a** Violin plots showing the percentage of variance in sex interaction estimates explained by gene expression; the dot represents the empirical variance, and the asterisk indicates the components that were significant against spatial null models. **b** Scatterplot of the association between the sex interaction estimates and the regional weights of the first and third **c** latent variables. Pearson's correlation coefficient (r) and corresponding *P*-value are reported. The red line shows the linear regression fit, and the shaded error band represents the 95% confidence interval (CI) of the fit. **d** Brain maps of the sex interaction coefficients and their regional weights for the first and third (**f**) latent variables. **e**, **g** The top ten molecular function terms from the Gene Ontology knowledge base that are enriched in the positively and negatively weighted gene sets associated with sex effect on cortical thickness in iRBD. Terms are ranked based on the normalized enrichment score; darker colored bars present significantly enriched terms after FDR correction. Brain maps were generated with Surflce[77]. FDR = false discovery rate; GSEA = gene set enrichment analysis; iRBD = isolated REM sleep behavior disorder; LV = latent variable; PLS = partial least square.

males[26,41,42]. In PD, vertex-based analyses have shown sex effects on cortical thickness and volume, particularly in regions overlapping those identified in our iRBD cohort, with males displaying greater cortical thinning and smaller volumes than females[41]. Males with PD also exhibit more severe motor impairment and higher prevalence of RBD symptoms compared to females[41]. Deformation-based morphometry and connectivity studies further support this showing that females with PD exhibit less cortical atrophy and less white matter connectivity disruption than males[43]. In early PD stages, females have increased cortical connectivity, correlating with more preserved motor function over time[42]. Similar findings have been reported in DLB, where volumetric analyses demonstrate that females exhibit less cortical gray matter loss than males[27], with differences particularly

pronounced at younger ages[27]. Given that iRBD represents a prodromal stage for PD and DLB, our findings suggest that similar sex-based neuroanatomical protective mechanisms may already be present in iRBD females, contributing to less severe and more restricted cortical thinning compared to males. However, it is important to keep in mind that in addition to differences in pathophysiological mechanisms of iRBD, other factors have been put forward to explain sex imbalance. First, the presence of a referral bias and differences in screening methods across studies have been reported[32]. Second, sociocultural and environmental influences such as sleep environment and co-sleeping dynamics can influence whether symptoms are observed and reported, potentially leading to differential detection between sexes[44,45]. Furthermore, emotional factors such as stress and

**Table 3 | Molecular functions enriched in regions with sex interaction for LV1**

| GO identifier | GO term | Gene set size | Number of leading edge IDs | Enrichment score | Normalized enrich-ment score | FDR P-value |
|---|---|---|---|---|---|---|
| **Positively weighted genes with sex interaction (females > males)** | | | | | | |
| GO:0003707 | nuclear steroid receptor activity | 20 | 8 | 0.65 | 1.95 | **0.027** |
| GO:0004984 | olfactory receptor activity | 48 | 38 | 0.45 | 1.69 | 0.13 |
| GO:0004386 | helicase activity | 138 | 46 | 0.36 | 1.62 | 0.17 |
| GO:0140993 | histone modifying activity | 187 | 61 | 0.33 | 1.61 | 0.16 |
| GO:0003727 | single-stranded RNA binding | 72 | 36 | 0.38 | 1.55 | 0.22 |
| **Negatively weighted genes with sex interaction (females < males)** | | | | | | |
| GO:0140104 | molecular carrier activity | 83 | 35 | −0.41 | −1.65 | 0.11 |
| GO:0016645 | oxidoreductase activity, acting on the CH-NH group of donors | 25 | 12 | −0.56 | −1.73 | 0.07 |
| GO:0043177 | organic acid binding | 145 | 54 | −0.41 | −1.75 | 0.07 |
| GO:0001664 | G protein-coupled receptor binding | 193 | 57 | −0.39 | −1.77 | 0.08 |
| GO:0005507 | copper ion binding | 45 | 18 | −0.51 | −1.78 | 0.10 |

Bold values represent significantly enriched gene terms. Only the top 10 gene terms are shown (redundancy reduction weight set cover applied).
FDR = false discovery rate; GO = gene ontology; LV = latent variable.

**Table 4 | Gene association within nuclear steroid receptor activity for LV1**

| Gene | Bootstrap ratio | Gene name |
|---|---|---|
| *ESRRG* | 13.33 | Estrogen-related receptor gamma |
| *ESRRA* | 10.92 | Estrogen-related receptor alpha |
| *NR3C2* | 9.07 | Nuclear receptor subfamily 3 group C member 2 |
| *NR2C1* | 7.57 | Nuclear receptor subfamily 2 group C member 1 |
| *NR3C1* | 7.31 | Nuclear receptor subfamily 3 group C member 1 |
| *PPARD* | 7.26 | Peroxisome proliferator activated receptor delta |
| *NR1D1* | 6.33 | Nuclear receptor subfamily 1 group D member 1 |
| *ESRRB* | 5.15 | Estrogen-related receptor beta |

Only showing the 8 significant gene terms included in this molecular function.
LV = Latent variable.

**Table 5 | Molecular functions enriched in regions with sex interaction for LV3**

| GO identifier | GO term | Gene set size | Number of leading edge IDs | Enrichment score | Normalized enrich-ment score | FDR P-value |
|---|---|---|---|---|---|---|
| **Positively weighted genes with sex interaction (females > males)** | | | | | | |
| GO:0004984 | olfactory receptor activity | 48 | 29 | 0.49 | 2.06 | **0.015** |
| GO:0003707 | nuclear steroid receptor activity | 20 | 8 | 0.54 | 1.80 | 0.09 |
| GO:0004386 | helicase activity | 138 | 53 | 0.33 | 1.67 | 0.11 |
| GO:0140993 | histone modifying activity | 187 | 53 | 0.27 | 1.46 | 0.27 |
| **Negatively weighted genes with sex interaction (females < males)** | | | | | | |
| GO:0003735 | structural constituent of ribosome | 156 | 65 | −0.37 | −1.60 | 0.10 |
| GO:0001664 | G protein-coupled receptor binding | 193 | 80 | −0.37 | −1.62 | 0.10 |
| GO:0016209 | antioxidant activity | 63 | 23 | −0.46 | −1.71 | 0.09 |
| GO:0043177 | organic acid binding | 145 | 51 | −0.41 | −1.75 | 0.09 |
| GO:0031681 | G-protein beta-subunit binding | 19 | 11 | −0.59 | −1.76 | 0.10 |
| GO:0030594 | neurotransmitter receptor activity | 71 | 26 | −0.47 | −1.82 | 0.08 |

Bold values represent significantly enriched gene terms. Only the top 10 gene terms are shown (redundancy reduction weight set cover applied).
FDR = false discovery rate; GO = gene ontology; LV = latent variable.

anxiety seem to impact dream-enacting behaviors differently in females and males with iRBD[46]. Finally, sex differences in polysomnography measures of sleep architecture (depending on hormonal status) may contribute to differences in how iRBD manifests and is captured across sexes[47].

Selective protection refers to the ability of specific brain regions to resist neurodegeneration, even in the presence of pathological processes[48]. In this study, only 1% of cortical regions in iRBD females showed significant atrophy, compared to 37% in males, despite both groups having similar age, clinical severity, and spatial patterns of cortical thinning Cortical thinning was particularly prominent in the occipital cortex, but also in parietal and dorsolateral prefrontal regions, which is consistent with previous reports in iRBD[7,8], PD[49], and DLB[27], suggesting overlapping neurodegenerative mechanisms. To

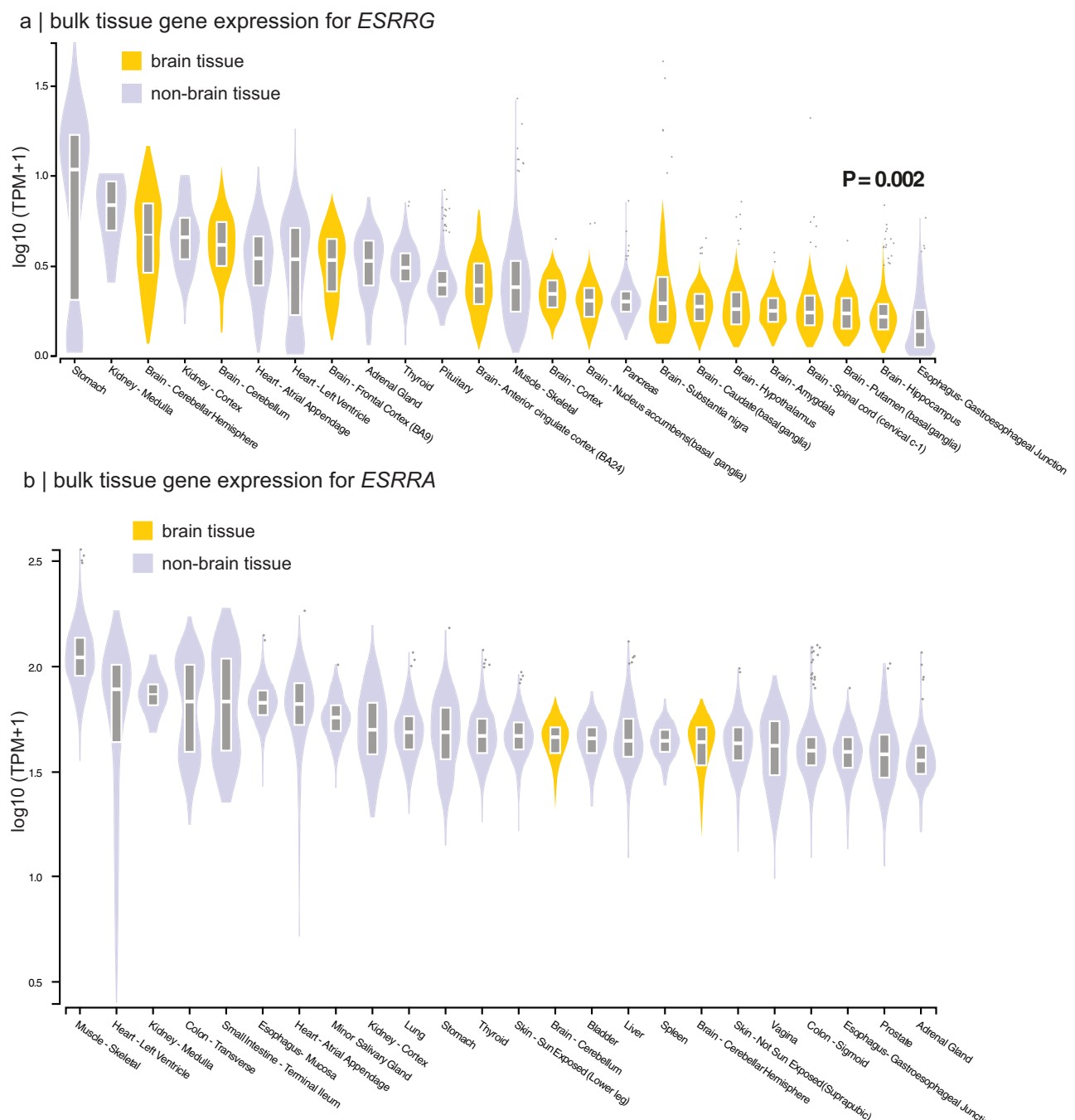

**Fig. 5 | Top 24 tissues expression of *ESRRG* and *ESRRA*. a, b** Violin plots of expression values of *ESRRG* and *ESRRA* across tissue types, with brain tissues shown in yellow and non-brain tissues shown in lilac. The *P*-value indicates a significantly higher proportion of brain tissues overexpressing *ESRRG* compared to *ESRRA*. Box plots display the median (center line) and the interquartile range (box; 25th-75th percentiles). Points appearing outside of the boxes correspond to outliers above or below 1.5 times the interquartile range. The expression of the top 24 tissue types, out of 54 available, are shown. Gene expression levels for each tissue were obtained from the GTEx Analysis Release V10 (dbGAP Accession phs000424.v10.p2) via the GTEx Portal (accessed 11/22/2024). *ESRRA* = estrogen-related receptor alpha; *ESRRG* = estrogen-related receptor gamma; GTEx = genotype-tissue expression; TPM = transcripts per million.

investigate the biological basis of selective protection in iRBD females, we analyzed gene expression patterns overlapping with regions showing sex-related differences in cortical thinning. Gene enrichment analysis identified nuclear steroid receptor functions, particularly estrogen-related receptors gamma and alpha (*ESRRG* and *ESRRA*) genes, as strongly associated with regions showing less atrophy in iRBD females. Notably, *ESRRG* exhibited more brain-specific over-expression. This contrasts with the more ubiquitous expression of *ESRRA*, underscoring the potential role of *ESRRG* in supporting selective brain protection. Interestingly, iRBD females had fewer years of education compared to iRBD males in this study. Given that higher education is associated with greater brain reserve and potentially brain resilience, this provides further support for the presence of protective factors being at play in the brain of iRBD females in comparison to males[50]. In this work, education levels were not controlled for since information about the number of years of education was not available for all sites. Future studies should investigate more thoroughly the impact of education on brain structure in iRBD.

Estrogen-related receptors gamma, alpha and beta (ERRγ, ERRα, ERRβ) are nuclear receptors involved in mitochondrial biogenesis, energy homeostasis, and neuronal metabolism[51]. ERRγ and ERRα act as co-regulators of peroxisome proliferator-activated receptor-gamma coactivator-1-alpha (PGC-1α), a key regulator of mitochondrial function, oxidative metabolism, and synaptic maintenance[51]. PGC-1α is highly expressed in regions vulnerable to neurodegeneration, including the cerebral cortex, hippocampus, striatum, and substantia nigra[52]. Disruptions in PGC-1α signaling and mitochondrial dysfunction are well-established contributors of synucleinopathy-related neurodegeneration, leading to energy deficits, oxidative stress, and impaired protein clearance[53]. Work from our group has shown that atrophy-prone regions in iRBD overexpress mitochondrial-related genes[16], supporting the idea that neuronal energy deficits play a role in iRBD-related brain neurodegeneration. Furthermore, recent findings indicate that *ESRRG* expression enhances mitochondrial function, protecting dopaminergic neurons from synuclein-induced toxicity and promoting protection against neurodegeneration[54]. Fox et al. demonstrated that deleting ERRγ in dopaminergic neurons increased vulnerability to alpha-synuclein toxicity, reduced mitochondrial gene expression, and decreased mitochondrial number, whereas ERRγ overexpression reduced alpha-synuclein aggregation and delayed neurodegeneration[54]. Further supporting this hypothesis, Ciron et al. demonstrated that PGC-1α deficiency exacerbates neurodegeneration in PD, leading to increased mitochondrial dysfunction and alpha-synuclein toxicity, with males being more vulnerable to neurodegeneration following PGC-1α loss[55]. Given that PGC-1α and ERRγ interact to regulate mitochondrial function, our findings suggest that *ESRRG* overexpression in atrophy-resistant cortical areas in iRBD females may contribute to mitochondrial protection, counteracting the neurodegenerative processes observed in males.

The neuroprotective potential of *ESRRG* is particularly compelling given our GTEx analysis, which revealed significant overexpression of *ESRRG* in the brain, in contrast to *ESRRA*, which is more ubiquitously expressed. This is in line with McMeekin et al. showing that disabling *ESRRA* does not significantly impact mitochondrial gene expression[56] suggesting that *ESRRG* may play a brain-specific role in neuroprotection, contributing to less neurodegeneration in iRBD females. Beyond *ESRRG* and *ESRRA*, another notable gene identified in atrophy-resistant regions of iRBD females was *PPARD*. PPARδ is a nuclear receptor involved in lipid metabolism, energy homeostasis, and mitochondrial function[57]. Among PPAR subtypes, PPARδ is the most highly expressed in the brain, where it regulates neuronal cell survival, neuroinflammation, and neurodegeneration resistance[57]. PPARδ has demonstrated neuroprotective effects in multiple neurodegenerative diseases, including PD[58] and Alzheimer's disease[59]. Preclinical and clinical studies suggest that PPARδ activation protects against mitochondrial dysfunction in PD models, reduces amyloid burden, and improves cognitive function in Alzheimer's disease[58,59]. The over-representation of *PPARD* in atrophy-resistant cortical regions of iRBD females, alongside *ESRRG*, suggests a broader network of transcription factors contributing to sex-based protection and highlights potential therapeutic targets for synucleinopathies.

Our analyses also revealed that genes associated with olfactory receptor activity were overexpressed in brain regions overlapping with those showing less atrophy in females with iRBD compared to males. Hyposmia is a well-established feature of synucleinopathies and is prevalent in iRBD[1,6]. Notably, sex differences in olfactory function are well documented, with females typically outperforming males in odor detection, identification, discrimination, and memory[60]. These differences persist with aging[61] with stronger electrophysiological responses and distinct activation patterns in olfactory brain regions in females[62,63]. Neuroanatomical studies also point to a sexually dimorphic olfactory system, likely influenced by sex hormones such as estrogens, which may enhance olfactory processing through increased neuronal sensitivity and synaptic modulation[64]. In this study, iRBD females tended to perform better on olfactory tests than males, although this difference did not reach statistical significance. This may be due to smaller female sample size and variability across the four olfactory testing scales used. Nevertheless, the observed transcriptomic and anatomical patterns may reflect a form of sex-specific protection in olfactory-associated regions, warranting further investigation.

This study has some limitations. First, while the multicentric dataset is a major strength, providing a large and diverse sample, the availability of detailed clinical and demographic data, including sex versus gender distinctions, as well as premorbid IQ, education levels of all participants (including controls) and hormonal status of females included in the study, was limited. Future studies should aim to disentangle these biological and sociocultural variables to better explore brain-clinical relationships in iRBD. Second, the male predominance in iRBD resulted in a smaller proportion of female participants. Despite this limitation, our study provides evidence of sex-related differences, making it the largest neuroimaging investigation of females with iRBD to date. Third, the gene expression data used in this study, obtained from the Allen Human Brain Atlas, provided high-resolution insights into over 20,000 genes across brain regions. these data were derived from healthy postmortem brains rather than iRBD-specific samples. Nevertheless, the ratios of males and females in the donors (83%) was equivalent to the one in our iRBD group (83%). Collecting postmortem gene expression data for every brain region remains a challenge, particularly for iRBD, where most patients convert to a neurodegenerative disease by the time their brains become available for study. Lastly, this study provides cross-sectional insights into sex differences in iRBD. Longitudinal follow-ups will be essential to understanding how sex-based differences influence phenoconversion to PD, DLB or MSA. As more patients progress to manifest synucleinopathies, future analyses will enable sex-stratified investigations into disease progression.

In summary, this study reveals significant sex-related differences in cortical atrophy in iRBD, with females showing less severe and less widespread cortical thinning than males. Gene enrichment analyses identified estrogen-related pathways, particularly the *ESRRG* gene, as potential contributors to selective protection in females. These findings provide insights into sex-based neuroprotection in prodromal synucleinopathies. This highlights potential directions for targeted therapeutic strategies, as well as the importance of sex in this line of research. Indeed, the pathways overexpressed in regions less affected in females may be amenable to pharmacological modulation, and any neuroprotective program built on these pathways should incorporate sex as a biological variable. Indeed, that iRBD females accumulate less cortical atrophy than clinically-matched males provides an argument against indiscriminate pooling of sexes in trials. Stratifying randomization may yield groups with more homogenous baseline burden and progression rates, thereby increasing statistical power and reducing sample size requirements. Finally, because normative cortical thickness distributions and atrophy trajectories differ by sex, quantitative MRI endpoints used to evaluate treatment efficacy should be scored against sex-specific reference curves. Taken together, sex is an important factor impacting neurodegeneration in iRBD patients.

## Methods
### Participants
A total of 888 participants were recruited for this study and underwent T1-weighted brain MRI imaging. This cohort included 408 polysomnography-confirmed iRBD patients and 480 age-matched healthy controls, recruited from nine international centers: 179 (85 patients) from the Center for Advanced Research on Sleep Medicine at the Hôpital du Sacré-Cœur de Montréal and The Neuro (controls from Quebec Parkinson Network), Montréal, Canada; 140 (83 patients) from the First Faculty of Medicine at Charles University, Prague, Czechia; 147

(81 patients) from the Oxford Discovery Cohort, Oxford, UK; 136 (60 patients) from the Movement Disorders Clinic (ICEBERG and ALICE cohorts) at the Hôpital de la Pitié-Salpêtrière, Paris, France; 56 (30 patients) from the ForeFront PD Research Clinic, Sydney, Australia; 38 (18 patients) from Aarhus University Hospital, Aarhus, Denmark; 29 (14 patients) from IRCCS Ospedale Policlinico San Martino, Genoa, Italy; and 163 (37 patients) from the Parkinson's Progression Markers Initiative (PPMI) baseline cohort.

All iRBD patients underwent video-polysomnography and were diagnosed based on the International Classification of Sleep Disorders criteria[2]. Neurological evaluations and cognitive assessments confirmed that patients were still in the isolated phase of RBD. Patients were excluded if they had, at the clinical visit closest in time to the MRI session, a diagnosis of DLB, PD or MSA based on published diagnostic criteria[65–67], had a history of brainstem stroke, epilepsy or epileptiform activity on EEG, had antidepressant-triggered RBD, had untreated obstructive sleep apnea, or had RBD mimics such as sleepwalking and night terrors. All patients underwent standardized clinical evaluations, including motor assessments using the Movement Disorders Society-sponsored Unified PD Rating Scale (MDS-UPDRS-III), global cognitive evaluation using the Montreal Cognitive Assessment (MoCA) and assessment of olfactory identification performance. Each cohort underwent either the 12-item Sniffin' Sticks, the 16-item Sniffin' Sticks, the 40-item University of Pennsylvania Smell Identification Test (UPSIT-40), or the reduced 12-item version of the UPSIT (UPSIT-12). For allowing comparisons between iRBD males and females, the 12-item Sniffin' Sticks, UPSIT-40, and UPSIT-12 scores were converted into a 16-item Sniffin' Sticks score following a previously developed calibration method[37]. The sex of participants was assigned by the clinician (female or male) during the clinical interview. All participants provided written informed consent. Participants included in this study were part of previous multicentric work investigating brain neurodegeneration in iRBD[8,12,16]. Study protocols were approved by the Research Ethics Board of the Quebec Integrated University Center for Health and Social Services of Northern Island of Montreal (MEO-37-2024-2699), the McGill University Health Center (MP-37-2022-7744), and the respective local ethics boards at all participating sites.

## MRI acquisition and processing
T1-weighted MRI scans were acquired using 3 T MRI scanners across the different sites, with site-specific acquisition protocols (detailed in the Supplementary Material). To assess sex-related effects on cortical morphology in iRBD, MRI scans were processed to generate whole-brain vertex-based cortical thickness maps using FreeSurfer (version 7.1.1). The standard processing pipeline included intensity normalization, brain extraction, segmentation of subcortical structures, cortical surface reconstruction, and topological correction (technical details in Supplementary Material). Cortical thickness was calculated as the closest distance between the gray-white and gray-CSF boundaries at each vertex, producing high-resolution maps sensitive to submillimeter cortical differences. All cortical reconstructions were visually inspected by trained raters (S.R. and V.A.) and scored based on established guidelines[36], as done previously[8,16]. Secondary analyses were conducted on cortical surface area and cortical volume maps, with surface area defined as the sum of triangle areas at each vertex and cortical volume as the product of cortical thickness and surface area. Additionally, subcortical volumes were extracted for all participants using FreeSurfer standard processing pipeline to investigate the presence of a sex-by-group interaction in subcortical regions. These regions were the left and right thalamus, caudate, putamen, pallidum, hippocampus, amygdala, and accumbens.

To characterize brain regions showing significant sex effects on cortical thickness and performing gene expression analysis, cortical maps were parcellated. We used the high-resolution Lausanne atlas, composed of 1000 cortical regions, to improve spatial granularity of

sex effects on atrophy[68]. The fetch_cammoun2012 function from netneurotools[68] was used to retrieve atlas annotation files in fsaverage5 template space, which were then registered to each participant's cortical surface via FreeSurfer's mri_surf2surf function. The aligned annotation files were converted to gifti format using mris_convert for compatibility with downstream processing, and cortical thickness measures were extracted for each parcellated region with custom scripts. To account for scanner-specific effects in this multicentric dataset, cortical thickness values were harmonized across acquisition sites using the ComBat tool[69], an empirical Bayes-based approach originally developed for genomics and widely applied in multicentric MRI studies, including our previous work[8,16,70]. ComBat correction was applied while preserving biological variability of interest (i.e., group, age, and sex), thereby removing variance attributable to site-specific effects. To ensure that age effects did not confound the atrophy-gene expression analyses, we applied W-scoring, a normalization technique used to remove age and sex effects in normal aging[8,16,26]. For each iRBD patient, W-scores were computed for all parcellated regions using linear regression models with the healthy control data, adjusting for age and sex. W-scores represent the standardized difference between observed and expected cortical thickness in iRBD patients, normalized by residual variance[26]. Negative W-scores indicate greater cortical thinning relative to age- and sex-matched controls. To quantify sex-related effects, we performed an additional set of linear regression analyses on the cortical thickness W-scores across parcellated regions. For each region, the ComBat-corrected W-scores were used as the dependent variable, with sex as the independent variable, producing a single beta coefficient per region. A positive beta indicated that iRBD females exhibited less cortical thinning than iRBD males, whereas a negative beta indicated greater cortical thinning in females. These sex interaction estimates were subsequently used as input in the imaging transcriptomics analysis.

## Gene expression extraction from postmortem brains
To investigate the gene expression patterns associated with sex-related cortical atrophy in iRBD, we extracted gene expression data from the Allen Human Brain Atlas (AHBA) in 1000 cortical regions (499 regions in the left hemisphere)[68,71]. The AHBA provides gene expression data for over 20,000 genes, quantified across 3200 tissue samples from six post-mortem healthy adult brains. Gene expression microarray data was accessed using abagen (version 0.1.3), following recommendations for preprocessing and normalization[14]. Probe-to-gene annotations were verified for accuracy, and probes failing to exceed background noise in at least 50% of samples across donors were discarded[72]. For genes associated with multiple probes, the probe showing the most stable expression across brain regions was selected. Tissue samples were mapped to the parcellated Lausanne atlas used for atrophy quantification by aligning MNI coordinates, ensuring hemisphere- and structure-specific correspondence (e.g., cortex vs. subcortex). Samples that could not be accurately assigned to a specific brain region were excluded. To mitigate variability across donors, gene expression values were normalized across samples, and regional expression values were averaged first within donors and then across the six donor brains, producing a regional gene expression matrix. Genes with inconsistent expression across donors were excluded. The AHBA includes right hemisphere samples from only two donors, and previous studies have shown minimal lateralization in microarray expression patterns[71]. Thus, current analyses were performed in the left hemisphere, as done previously[16].

## Imaging transcriptomics
We applied partial least squares (PLS) regression[73] to determine whether regional gene expression patterns were associated with sex effects on cortical atrophy in iRBD. PLS regression enabled the identification of latent variables maximizing covariance between the

regional cortical thickness sex interaction estimates (for the 499 regions in the left hemisphere) and gene expression levels (15,633 genes across the same 499 regions). Given the high spatial auto-correlation inherent in the brain[74], we ensured that gene-atrophy associations were not driven by lower-order spatial gradients by comparing the empirical variance explained by each latent variable to 10,000 spatially constrained null models. Brain regions were randomly shuffled using a spherical reassignment procedure that preserved spatial autocorrelation[75]. A latent variable was considered significant if fewer than 5% of null models explained more variance than the original atrophy vector. To identify the genes most strongly associated with significant latent variables, we applied a bootstrapping resampling procedure. Rows of the gene expression and cortical thickness matrices were randomly shuffled, and PLS regression was repeated on these shuffled matrices. This procedure was iterated 5000 times, generating a null distribution and standard errors for each gene's weight. Boot-strap ratios (the ratio of each gene's weight to its bootstrap-estimated standard error) were interpreted as z-scores, allowing genes to be ranked from highest to lowest based on their bootstrap ratios. These ranked gene lists were subsequently used as inputs for gene set enrichment analysis.

### Gene set enrichment analysis

To identify enriched functional components in the genes associated with sex-specific cortical atrophy in iRBD, we performed gene set enrichment analysis using WebGestalt[76]. This analysis aimed to uncover molecular functions, biological processes, and cellular com-ponents associated with regional sex effects on cortical thinning. Gene terms containing a minimum of five and a maximum of 2000 genes were included in the enrichment analysis. To correct for multiple comparisons, 1000 random permutations were conducted, with P-values adjusted using the false discovery rate (FDR) method. Sig-nificant non-redundant terms associated with less atrophy in iRBD females compared to males (relative to age-matched controls) were interpreted. This approach has shown efficacy and specificity in neu-rodegenerative diseases, with atrophy-gene correlations in synuclei-nopathies highlighting mitochondrial and macroautophagy functions[16], protein modeling complexes (including *APOE*) in Alzhei-mer's disease[16], and oligodendrocytic cell types in MSA[18]. To ensure that identified gene terms were not biased by the choice of a specific gene ontology platform, we performed a complementary enrichment analysis using GOrilla[39].

### Brain vs. peripheral gene expression

To explore whether genes implicated in sex-related cortical atrophy in iRBD were selectively expressed in the central nervous system (specific to brain neurodegeneration) or extended to peripheral tis-sues, we analyzed gene expression patterns using data from the GTEx Project[40]. GTEx provides a comprehensive resource for studying human gene expression across 54 non-diseased tissue types, col-lected from nearly 1000 post-mortem donors, enabling a detailed assessment of gene expression distribution throughout the body. Gene expression levels for each tissue were obtained from the GTEx Analysis Release V10 (dbGAP Accession phs000424.v10.p2) via the GTEx Portal (accessed 11/22/2024). Expression was quantified using bulk RNA sequencing, reported as transcripts per million (TPM), with isoforms collapsed into a single gene model. For each gene, tissues were classified as overexpressing the gene if their expression exceeded the median expression value across all tissues. To assess whether these genes were disproportionately expressed in the brain compared to the rest of the body, we performed chi-squared tests for each gene, comparing the proportion of overexpressing brain tissues to all other tissue types. Only tissues with available gene expression data for all analyzed genes were included, resulting in 54 distinct tissue types examined.

### Statistical analysis

Group differences in continuous demographic and clinical variables were assessed using independent sample t-tests for normally dis-tributed variables and Mann-Whitney U test for non-normally dis-tributed variables. Effect sizes were measured as Cohen's d. Categorical variables were compared using chi-squared tests. Vertex-based cortical thickness analyses were conducted in FreeSurfer using general linear models to assess sex-by-group interactions on cortical thickness, surface area, and volume, adjusting for age and acquisition site. For surface area and cortical volume analyses, estimated total intracranial volume (eTIV) was included as an additional covariate. For the analysis of subcortical volumes, volumes were also normalized by the eTIV. Analyses were performed separately for the left and right hemispheres, with P-values adjusted for both hemispheres using the 2spaces flag. Surface maps were smoothed with a 15-mm full-width half maximum kernel. Additional vertex-based general linear models were performed to examine sex differences within each group separately. Cluster-level significance was determined using Monte Carlo spatial permutations, with a statistical threshold of $P < 0.05$ and vertex-level significance set at $P < 0.05$. We performed additional analyses to test the robustness of our identified clusters, using cluster-forming thresholds of $P < 0.01$. For subcortical volume analyses, a statistical threshold of $P_{FDR} < 0.05$ was used to determine significance. To vali-date our results with a parcel-wise approach, we extracted region-based cortical thickness values from the 1000 cortical region Lausanne parcellation across hemispheres. One-sample t tests were conducted to compare ComBat-corrected W-scored cortical thickness values of male and female iRBD patients to 0, representing the mean cortical thickness in healthy controls. FDR correction was applied to control for multiple comparisons. Correlations between continuous variables were assessed using Pearson's correlation coefficient (r). All statistical analyses were performed using SPSS, R, Python, and MATLAB.

### Reporting summary

Further information on research design is available in the Nature Portfolio Reporting Summary linked to this article.

## Data availability

The imaging and clinical data used in this study were obtained from multiple collaborating centers, each of which retains ownership of their respective datasets. The principal investigator had authorized access to all data necessary for the analyses performed in this study. Access to the data is restricted due to institutional policies and parti-cipant privacy regulations. Requests for data access must be submitted directly to the respective data-holding institutions and are subject to their local ethical and legal frameworks. These restrictions limit the sharing of raw data in a public repository. Access requests can be made directly to each collaborating center, each with their own require-ments. Source data generated and analyzed during this study, where permitted, are provided with this paper. Source data are provided with this paper.

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

## Acknowledgements

The authors thank all participants included in the different cohorts studied. This study was supported by grants to S.R. from Alzheimer Society Canada (0000000082) and Parkinson Canada (PPG-2023-0000000122). This work was also supported by a donation in memoriam of Gaetan Boulianne and Pierre-Claude Durivage to the Foundation of Hopital du Sacré-Coeur de Montréal, in support of PD research at the Center for Advanced Research in Sleep Medicine. M.F. holds an excellence scholarship from the Faculty of Medicine of University of Montreal and the Solange-Saul excellence scholarship. S.R. reports a Junior 1 research scholar award from Fonds de recherche du Québec – Santé. The work performed in Montreal was supported by the Canadian Institutes of Health Research (CIHR), the Fonds de recherche du Québec – Santé (FRQ-S), the W. Garfield Weston Foundation, and F. Hoffmann-La Roche AG. The work performed in Oxford was funded by Parkinson's UK (J-2101) and the National Institute for Health Research (NIHR) Oxford Biomedical Research Center (BRC). The work performed in Prague was funded by the Czech Health Research Council grant NU21-04-00535 and by project nr. LX22NPO5107 (MEYS): Financed by European Union – Next Generation EU. The work performed in Paris was funded by grants from the Program d'investissements d'avenir (ANR-10-IAIHU-06), the Paris Institute of Neurosciences – IHU (IAIHU-06), the Agence Nationale de la Recherche (ANR-11-INBS-0006), Électricité de France (Fondation d'Entreprise EDF), Control-PD (Joint Program–Neurodegenerative Disease Research [JPND] Cognitive Propagation in Prodromal PD), the Fondation Thérèse et René Planiol, the Fonds Saint-Michel; by unrestricted support for research on PD from Energipole (M. Mallart) and Société Française de Médecine Esthétique (M. Legrand); and by a grant from the Institut de France to Isabelle Arnulf (for the ALICE Study). The work performed in Sydney was supported by a Dementia Team Grant from the National Health and Medical Research Council (#1095127). The work performed in Aarhus was supported by funding from the Lundbeck Foundation, Parkinsonforeningen (The Danish Parkinson Association), and the Jascha Foundation. The work performed as part of the Parkinson's Progression Markers Initiative (PPMI), a public-private partnership, was funded by the Michael J. Fox Foundation for Parkinson's Research and funding partners, including 4D Pharma, AbbVie Inc., AcureX Therapeutics, Allergan, Amathus Therapeutics, Aligning Science Across Parkinson's (ASAP), Avid Radiopharmaceuticals, Bial Biotech, Biogen, BioLegend, Bristol Myers Squibb, Calico Life Sciences LLC, Celgene Corporation, DaCapo Brainscience, Denali Therapeutics, The Edmond J. Safra Foundation, Eli Lilly and Company, GE Healthcare, GlaxoSmithKline, Golub Capital, Handl Therapeutics, Insitro, Janssen Pharmaceuticals, Lundbeck, Merck & Co., Inc., Meso Scale Diagnostics, LLC, Neurocrine Biosciences, Pfizer Inc., Piramal Imaging, Prevail Therapeutics, F. Hoffman-La Roche Ltd and its affiliated company Genentech Inc., Sanofi Genzyme, Servier, Takeda Pharmaceutical Company, Teva

Neuroscience, Inc., UCB, Vanqua Bio, Verily Life Sciences, Voyager Therapeutics, Inc., and Yumanity Therapeutics, Inc. For up-to-date information on the study, visit www.ppmi-info.org.

## Author contributions

M.F. contributed in data analysis and draft writing; V.A., C.T., C.H., V.D., A.P.B., J.F.G., R.B.P., P.De, S.M., Z.V., J.K., M.T.H., I.A., P.Do, M.V., J.C.C., S.L., S.L., E.M., K.A.E.M., L.C., P.B., K.K., A.K.H., D.A., B.O., P.M., and L.R. contributed in data collection and manuscript revision; S.R. contributed in data analysis, draft writing and study supervision.

## Competing interests

There are no competing interests.

## Additional information

[1]Centre for Advanced Research in Sleep Medicine, Hôpital du Sacré-Cœur de Montréal, Quebec Integrated University Centre for Health and Social Services of Northern Island of Montreal, Montreal, QC, Canada. [2]Department of Neuroscience, University of Montreal, Montreal, QC, Canada. [3]Department of Psychology, University of Montreal, Montreal, QC, Canada. [4]Department of Psychology, Université du Québec à Montréal, Montreal, QC, Canada. [5]Research Centre, Institut universitaire de gériatrie de Montréal, Montreal, QC, Canada. [6]The Neuro (Montreal Neurological Institute-Hospital), McGill University, Montreal, CanadaQC. [7]Department of Neurology, Montreal General Hospital, Montreal, QC, Canada. [8]Department of Neurology and Centre of Clinical Neurosciences, First Faculty of Medicine, Charles University and General University Hospital, Prague, Czechia. [9]Oxford Parkinson's Disease Centre and Division of Neurology, Nuffield Department of Clinical Neurosciences, University of Oxford, Oxford, UK. [10]Sorbonne University, Institut du Cerveau (Paris Brain Institute, ICM), Inserm, CNRS, Assistance Publique Hôpitaux de Paris, Paris, France. [11]Parkinson's Disease Research Clinic, Macquarie Medical School, Macquarie University, Sydney, NSW, Australia. [12]Faculty of Medicine and Health, University of Sydney, Camperdown, NSW, Australia. [13]Department of Kinesiology and Health Sciences, University of Waterloo, Waterloo, ON, Canada. [14]Department of Nuclear Medicine and PET, Aarhus University Hospital, Aarhus, Denmark. [15]Department of Neuroscience, Rehabilitation, Ophthalmology, Genetics, Maternal and Child Health (DINOGMI), Clinical Neurology, University of Genoa, Genoa, Italy. [16]IRCCS Ospedale Policlinico San Martino, Genoa, Italy. [17]Department of Health Sciences, University of Genova, Genova, Italy. [18]Department of Medicine, University of Montreal, Montreal, QC, Canada. ✉e-mail: shady.rahayel@umontreal.ca

## ICEBERG Study Group

**Steering committee** Marie Vidailhet[10], Jean-Christophe Corvol [10], Isabelle Arnulf [10], Stéphane Lehéricy[10]

**Clinical data** Marie Vidailhet[10], Jean-Christophe Corvol [10], Isabelle Arnulf [10]

**Sleep assessment** Isabelle Arnulf [10], Pauline Dodet [10]

**Genetic data** Jean-Christophe Corvol [10]

**Brain MRI data** Stéphane Lehéricy[10]

