## [Transparent Peer Review file · Nature Communications]

Estrogen-related receptor gene expression associates with sex differences in cortical atrophy in isolated REM sleep behavior disorder

Corresponding Author: Ms Marie Filiatrault

Version 0:

Reviewer comments:

Reviewer #1

(Remarks to the Author)

This study investigated the sex differences in cortical thickness in 408 patients with iRBD. Across analyses, women with iRBD showed less atrophy than men, while no sex differences appeared in controls. Regional expression of estrogen-related genes was related to less atrophy in women.

The study investigates a relevant question of the field and is soundly executed, using a large cohort, variety of advanced analysis methods, a high-resolution brain atlas, comparison of gene expression between brain regions and body, and various sensitivity analyses. Additionally, the manuscript is well-written and, despite the complexity of analyses, easily comprehensible.

My main concern is that the study only assessed cortical atrophy while excluding subcortical regions. Previous studies showed differences in subcortical volumes in women and men with PD (Oltra et al., 2022, Front Aging Neurosci) and this difference might already arise in prodromal stages such as iRBD. Could you explain why subcortical atrophy was not evaluated in this study?

Additionally, I wonder about the following and would appreciate an explanation: Gene expression (including expression of estrogen-related genes) was extracted from the AHBA, which mainly includes male donors. If expression of estrogen-related genes is comparable in women and men across brain regions, I wonder which aspect of the expression product makes it protective in women but not in men?

I would appreciate a (supplementary) figure showing the gene expression of the most strongly associated genes across the brain to better see regional variability (ESRRG, ESRRa, PPARA).

Annegret Habich

Reviewer #2

(Remarks to the Author)

The authors provide results of utmost importance in the field of sex differences in early neurodegeneration stages. Among their findings, they report increased cortical atrophy in male iRBD patients relative to their female counterparts. Furthermore, they show that this differential cortical atrophy occurs in brain regions with high expression of steroid receptors, thereby suggesting a potential neuroprotective role of estrogens in brain degeneration in early stages. The work is noteworthy since it agrees with previous results in the field of alpha-synucleinopathies study showing sex differences in prodromal stages of the disease and highlights the importance of sex-difference findings in the era of personalised medicine. Moreover, genetic analyses give some insight on possible mechanisms underneath the sex-differences in the observed cortical neurodegeneration. Overall, the work should be commended for their large sample size, innovative approach and study design.

We include the following suggestions that may help improve the quality of the work:

1. Introduction

- The authors state that “iRBD is the strongest known prodromal marker of neurodegenerative synucleinopathies, with over 90% of individuals eventually developing...”.

Whereas the study by Galbiati is frequently cited, phenoconversion rates are slightly over 70% in other studies (Postuma et al., 2019, *Brain*; Zhang et al., 2022, *Annals of Neurology*).

- The authors state that “iRBD has a strong male predominance, with reported prevalence ratios as high as 8:1, leading to an underrepresentation of female individuals in iRBD studies, particularly in neuroimaging studies”.

Whereas a male predominance in iRBD has been observed, it remains unclear whether this is due to a sex difference in pathophysiological mechanisms of RBD, a referral bias or due to differences in screening methods across studies (Li, X., Zong, Q., Liu, L., Liu, Y., Shen, Y., Tang, X., Wing, Y. K., Li, S. X., & Zhou, J. (2023). Sex differences in rapid eye movement sleep behavior disorder: A systematic review and meta-analysis. *Sleep medicine reviews*, 71, 101810.

<https://doi.org/10.1016/j.smr.2023.101810>). We believe that, if not mentioned in the introduction, the discussion section would benefit from a small, although relevant mention to other variables potentially influencing an easier detection of iRBD in male participants, such as emotional and environmental factors affecting RBD manifestations (Jun et al., 2022, *Nature and science of sleep*, <https://doi.org/10.2147/NSS.S372823>), sex-related differential manifestation of motor events in RBD, may contribute to underdiagnosis in women (Bugalho et al. 2019, *Journal of clinical sleep medicine*, <https://doi.org/10.5664/jcsm.8086>) and sex differences in physiological sleep that may influence a more readily detection of RBD in males by their female counterparts (Kocevska, D. et al., 2021, *Nat Hum Behav*, <https://doi.org/10.1038/s41562-020-00965-x>; Nowakowski, S. et al. 2013, *Sleep and Women's Health*, <https://doi.org/10.17241/smr.2013.4.1.1>).

- The authors made a precise introduction regarding sex-differences in neurodegeneration. However, it will benefit from a justification of the hypothesis regarding the overexpressing genes involved in estrogen-related molecular functions and the suitability of using imaging transcriptomics to investigate spatial distribution of gene expression in the brain to underlies sex-related atrophy differences.

- The authors should state at the end of the introduction both the number of subjects included in the dataset (888) as the final number of subjects included in the analyses (687).

2. Methods section:

- The authors report that “all patients were part of ongoing prospective studies”. If the sample partially overlaps with that of previously published studies, these should be cited.

- In the statistical analyses section of the manuscript, it is mentioned that t-tests were used. Was normality of the variables tested? T-tests assume normal distributions and non-parametric tests must be used if normality is not met. This is particularly relevant for expected non-normality in variables such as the UPDRS-III.

- Cluster-level significance was determined using Monte Carlo spatial permutations, with a statistical threshold of $P < 0.05$ and vertex-level significance set at $P < 0.05$. , the cluster-forming threshold of 1.3 is quite liberal and increase type I error. Cluster-forming threshold of 2.0 or higher would be more appropriate.

3. Results:

- Of the 888 participants, 201 (23%) did not pass the quality control, resulting in 687 participants for analysis, please include a flow chart to specify the exclusion criteria. Additionally, the characteristics of the participants that were removed should also be reported, which was the proportion of male and female in the excluded sample?

- The authors reported the sex-difference between the iRBD groups despite similar age and clinical severity. Nevertheless, we believe that the presented clinical data does not allow to support this statement and that a more extensive clinical description of the iRBD sample is needed to ensure that clinical severity is comparable: current treatments, time since iRBD diagnosis (since PSG assessment) or time since beginning of symptomatology (since self-reported initiation of iRBD symptoms) should be included. Furthermore, any olfactory assessment available for the present sample? Please, report even if it is only available for a subgroup of the study sample. Sex differences in the available sample of olfactory data should be tested to rule out the potential influence of olfactory alterations in the reported sex differences in cortical thickness. If no olfactory assessment was performed, this should be stated as a limitation and authors should preclude statements assuring comparability of clinical symptoms between groups.

- In the line of our prior comment, inclusion of variables highly relevant in terms of brain resilience as education and premorbid IQ should be reported to guarantee the groups are comparable and the results interpretable.

- The iRBD sample did not differed significantly regarding the mean MoCA values. However, means and standard deviations in MoCA scores in the iRBD sample suggest the presence of individuals with MCI. Providing minimum and maximum MoCA scores for each study group would contribute to a better sample characterization, as well as the MoCA cut-off that would be used to determine MCI. It would be interesting to know the percentage of MCI participants in the male and female groups, both in the iRBD and the HC. Regarding this point, the between group differences in MOCA in HC group showed a trend of significance showing lower global cognition in male participants could the results have influenced by a low-level control group for males or a higher level in the case of female-group?

- To avoid false positive results, the authors should only report significant results after FDR correction, therefore the results sections should be modified accordingly including the text and figures. Moreover, the Effect size should be included in each analysis.

- In the result section, the authors state “However, the spatial pattern of atrophy is largely similar between sex, suggesting the presence of sex-specific protective mechanisms that may mitigate brain neurodegeneration in females”. This is an interpretation of findings and should not therefore belong to Results; move to discussion.

• Discussion:

- A relevant finding of the study regards the fact that the most strongly associated gene terms were enriched in olfactory receptor activity, although this finding is not further discussed. Given the fact that anosmia is a recognized non-motor prodromal symptom in alpha-synucleinopathies and prevalent in iRBD, this finding warrants further discussion.

- In the discussion of their results, the authors refer to resilience in the case of females and associate the higher resilience in

the group of females with estrogen gene expression in the regions of interest. Caution should be taken since the reported analyses do not control for highly relevant variables in terms of brain resilience, such as education and premorbid IQ. Additionally, this may actually differ between sexes.

- The limitations should include the lack of information regarding the hormonal status of the studied sample and its relationship with age.
- Implications of having used expression gene atlases mirroring left-hemisphere data. There is evidence suggesting lateralization of neurotransmission systems that are relevant for iRBD and related conditions (i.e. noradrenaline)

Minor:

- "and resistance to neurodegeneration resistance" (page 22). Remove one of "resistance".
- Figure 2. Correct typo "the range of t-values for each regions".
- only 4 regions (1%) remained significant after FDR correction, with t-scores ranging from -3.7 to -4.8 (Figure 2A). t-scores are shown in 2B, no?
- These findings reveal that cortical atrophy in iRBD is markedly less widespread and less severe in females compared to males ($\chi^2 = 210.65$, $P < 0.0001$), despite comparable age and clinical severity. Correct and write "is".
- Statistics that are reported in tables should not be also reported in the text.
- Table S1. Were differences corrected for multiple comparisons? This should be done and stated in each table.
- PGC and other acronyms should be fully written the first time they're mentioned

Reviewer #3

(Remarks to the Author)

Reviewer #4

(Remarks to the Author)

The topic is important. The work is original and the methodology is sound. The results are really interesting and the work supports the conclusions. There are some limitations, as reported by the same authors.

I only suggest shortening the Discussion section and avoiding some redundancies. However, the part concerning "The findings provide mechanistic insight of sex-specific neuroprotection in prodromal stages of synucleinopathies, potentially leading to novel targeted therapeutic approaches" could be extended.

Version 1:

Reviewer comments:

Reviewer #1

(Remarks to the Author)

I thank the authors for addressing all of my previous comments and questions.

Annegret (Greta) Habich

Reviewer #2

(Remarks to the Author)

Thank you to the authors for addressing the majority of my previous comments. There are, however, a few minor aspects that should still be included in the revised manuscript.

In line with my previous comments (11 and 16), the authors should explicitly mention as a limitation the lack of information regarding the educational level of the control group. Additionally, they should include in the Discussion their rationale for not controlling for education in the RBD between-group analyses—despite having the data to do so. Specifically, they should clarify that they expected iRBD males to show less neurodegeneration than females based on the observation that RBD males had higher education levels than females.

Furthermore, to enhance clarity, the authors should specify in both the Methods and Results sections the p-value threshold used to determine significance in the subcortical volumetric analyses.

Reviewer #3

(Remarks to the Author)

Reviewer #4

(Remarks to the Author)

The manuscript has been improved.

I have no further comments.

REVIEWER #1

1. This study investigated the sex differences in cortical thickness in 408 patients with iRBD. Across analyses, women with iRBD showed less atrophy than men, while no sex differences appeared in controls. Regional expression of estrogen-related genes was related to less atrophy in women. The study investigates a relevant question of the field and is soundly executed, using a large cohort, variety of advanced analysis methods, a high-resolution brain atlas, comparison of gene expression between brain regions and body, and various sensitivity analyses. Additionally, the manuscript is well-written and, despite the complexity of analyses, easily comprehensible.

We thank the reviewer for the positive comments.

2. My main concern is that the study only assessed cortical atrophy while excluding subcortical regions. Previous studies showed differences in subcortical volumes in women and men with PD (Oltra et al., 2022, Front Aging Neurosci) and this difference might already arise in prodromal stages such as iRBD. Could you explain why subcortical atrophy was not evaluated in this study?

We agree with the reviewer that subcortical structures are important in this disorder. We extracted the subcortical volumes from all participants and assessed the interaction between sex and group (iRBD patients vs controls) on subcortical volumes (normalized for head size), while controlling for age and acquisition site. We found no significant sex-by-group interaction in any of the subcortical regions ($P > 0.1$):

Table S4. Results of sex-by-group interaction analyses in subcortical volumes.

Subcortical region	Interaction coefficient^a	P-value^a
Left Thalamus	0.00020629	0.22496239
Left Caudate	5.76E-05	0.52305721
Left Putamen	0.00012337	0.3548939
Left Pallidum	-6.40E-06	0.9090655
Left Hippocampus	6.44E-05	0.52125199
Left Amygdala	-8.26E-06	0.85879768
Left Accumbens	1.52E-06	0.9402463
Right Thalamus	0.00033422	0.03957345
Right Caudate	1.18E-05	0.90142558
Right Putamen	0.0001429	0.29704106

Right Pallidum	-1.38E-05	0.79671207
Right Hippocampus	4.58E-05	0.64391585
Right Amygdala	-6.20E-05	0.14583868
Right Accumbens	3.15E-06	0.86524764

All analyses were done on subcortical volumes normalized for estimated total intracranial volume.

^a Linear regression - interaction model. Coefficient (β) and associated P-value.

We have modified the manuscript to mention that we did not find any differences in subcortical structures. The methods section now includes: “Additionally, subcortical volumes were extracted for all participants using FreeSurfer standard processing pipeline to investigate the presence of a sex-by-group interaction in subcortical regions. These regions were the left and right thalamus, caudate, putamen, pallidum, hippocampus, amygdala, and accumbens.” (pages 8-9, lines 348-357) and “For the analysis of subcortical volumes, volumes were also normalized by the eTIV.” (page 13, line 497). The results section “We also investigated the presence of a sex-by-group interaction on subcortical volumes and found no significant effect (Table S4).” (page 17, lines 599-601).

3. Additionally, I wonder about the following and would appreciate an explanation: Gene expression (including expression of estrogen-related genes) was extracted from the AHBA, which mainly includes male donors. If expression of estrogen-related genes is comparable in women and men across brain regions, I wonder which aspect of the expression product makes it protective in women but not in men?

We thank the reviewer for this important comment. The gene expression of estrogen-related receptors gamma (*ESRRG*) and alpha (*ESRRA*) was strongly correlated in female and male donors from the Allen Human Brain Atlas (AHBA) (*ESRRA*: $R = 0.72$, $P = 2.90e-165$; *ESRRG*: $R = 0.81$, $P = 2.63e-232$). However, that expression of these genes correlates in males and females does not mean that their effects are functionally equivalent, particularly in the context of neurodegeneration. Gene expression levels reflect the presence of a transcript, not the broader molecular context in which the gene product operates. Estrogen-related genes like *ESRRA* and *ESRRG* are involved in regulating mitochondrial function and oxidative phosphorylation, which are pathways known to interact with estrogen signaling (Sadavisam et al., 2024). Women have a distinct lifelong hormonal environment that can modulate the function of *ESRRA* and *ESRRG* downstream pathways, enhancing mitochondrial resilience and reducing neurodegenerative vulnerability. The title of the manuscript “Estrogen-related receptor genes underlie sex differences in cortical atrophy associated with isolated REM sleep behavior disorder” was thought out in this regard: it does not refer to differential gene expression between sexes but to the differential impact on neurodegeneration associated with these genes in sex-specific contexts.

Reference:

Sadasivam, N., Park, W.-R., Choi, B., Seok Jung, Y., Choi, H.-S., & Kim, D.-K. Exploring the impact of estrogen-related receptor gamma on metabolism and disease. *Steroids*. 2024; 211:109500.

4. I would appreciate a (supplementary) figure showing the gene expression of the most strongly associated genes across the brain to better see regional variability (*ESRRG*, *ESRRA*, *PPARD*).

We thank the reviewer for the comment. We added a figure in supplementary materials showing the average expression of *ESRRG*, *ESRRA*, and *PPARD* across donors of the AHBA and in males vs female separately, including their correlation (Figure S5).

We have added the following in the Results section: “Regional gene expression patterns for the whole group of donors, included separately by sex, are shown in Figure S5” (page 19, lines 667-668).

Figure S5. Regional gene expression patterns of *ESRRA*, *ESRRG*, and *PPARD*.

Regional cortical expression of *ESRRA* (a), *ESRRG* (b), and *PPARD* (c). For each gene, maps show all donors (left), females and males (middle), and accompanying scatterplots plotting male against female regional expression.

REVIEWER #2

1. The authors provide results of utmost importance in the field of sex differences in early neurodegeneration stages. Among their findings, they report increased cortical atrophy in male iRBD patients relative to their female counterparts. Furthermore, they show that this differential cortical atrophy occurs in brain regions with high expression of steroid receptors, thereby suggesting a potential neuroprotective role of estrogens in brain degeneration in early stages. The work is noteworthy since agree with previous results in the field of alpha-synucleinopathies study showing sex differences in prodromal stages of the disease and highlights the importance of sex-difference findings in the era of personalise medicine. Moreover, genetic analyses give some insight on possible mechanisms underneath the sex-differences in the observed cortical neurodegeneration. Overall, the work should be commended for their large sample size, innovative approach and study design.

We thank the reviewer for the positive comments.

2. We include the following suggestions that may help improve the quality of the work: Introduction. The authors state that “iRBD is the strongest known prodromal marker of neurodegenerative synucleinopathies, with over 90% of individuals eventually developing...”. Whereas the study by Galbiati is frequently cited, phenoconversion rates are slightly over 70% in other studies (Postuma et al., 2019, Brain; Zhang et al., 2022, Annals of Neurology).

We thank the reviewer for raising this. The mentioned information was changed on line 76, and references were adjusted as well. The introduction now reads as: “iRBD is the strongest known prodromal marker of neurodegenerative synucleinopathies, with over 70% of individuals eventually developing dementia with Lewy bodies (DLB), Parkinson’s disease (PD), or in a smaller proportion, multiple system atrophy (MSA).³⁻⁵”

References:

3. Galbiati, A., Verga, L., Giora, E., Zucconi, M. & Ferini-Strambi, L. The risk of neurodegeneration in REM sleep behavior disorder: A systematic review and meta-analysis of longitudinal studies. *Sleep Med. Rev.* 2019;43: 37–46.

4. Postuma, R. B. et al. Risk and predictors of dementia and parkinsonism in idiopathic REM sleep behaviour disorder: a multicentre study. *Brain.* 2019; 142:744–759.

5. Zhang, H. et al. Risk Factors for Phenoconversion in Rapid Eye Movement Sleep Behavior Disorder. *Ann. Neurol.* 2022; 91: 404–416.

3. The authors state that “iRBD has a strong male predominance, with reported prevalence ratios as high as 8:1, leading to an underrepresentation of female individuals in iRBD studies, particularly in neuroimaging studies”.

Whereas a male predominance in iRBD has been observed, it remains unclear whether this is due to a sex difference in pathophysiological mechanisms of RBD, a referral bias or due to differences in screening methods across studies (Li, X., Zong, Q., Liu, L., Liu, Y., Shen, Y., Tang, X., Wing, Y. K., Li, S. X., & Zhou, J. (2023). Sex differences in rapid eye movement sleep behavior disorder: A systematic review and meta-analysis. *Sleep medicine reviews*, 71, 101810. <https://doi.org/10.1016/j.smrv.2023.101810>). We believe that, if not mentioned in the introduction, the discussion section would benefit from a small, although relevant mention to other variables potentially influencing an easier detection of iRBD in male participants, such as emotional and environmental factors affecting RBD manifestations (Jun et al., 2022, Nature and science of sleep,

<https://doi.org/10.2147/NSS.S372823>), sex-related differential manifestation of motor events in RBD, may contribute to underdiagnosis in women (Bugalho et al. 2019, *Journal of clinical sleep medicine*, <https://doi.org/10.5664/jcsm.8086>) and sex differences in physiological sleep that may influence a more readily detection of RBD in males by their female counterparts (Kocevska, D. et al., 2021, *Nat Hum Behav*, <https://doi.org/10.1038/s41562-020-00965-x>; Nowakowski, S. et al. 2013, *Sleep and Women's Health*, <https://doi.org/10.17241/smr.2013.4.1.1>).

We thank the reviewer for this insightful comment and for pointing us toward relevant literature. In our revised manuscript, we now strengthen more thoroughly our discussion of the male predominance found in iRBD. As suggested, we have incorporated a more balanced discussion that includes, in addition to differential pathological mechanisms:

- Possible referral and detection bias, where iRBD may be more easily detected in males due to more violent or disruptive dream enactment behaviors, which are more likely to prompt medical consultation (Jun et al., 2022). These behaviors may be less pronounced or expressed differently in females, leading to underdiagnosis.
- Possible sex-related differences in motor symptom expression, where motor manifestation during REM sleep may potentially be subtler or less disruptive in women compared to men (Bugalho et al. (2019), making iRBD harder to detect.
- Possible sociocultural and environmental influences, where the sleep environment and co-sleeping dynamics can influence whether symptoms are observed and reported, potentially leading to differential detection between sexes (Kocevska et al., 2021; Nowakowski et al., 2013).
- Possible differences in physiological sleep architecture, where variations in REM sleep patterns between sexes may influence both the frequency and detectability of abnormal behaviors (Lok et al., 2024).

We have revised the Discussion section to incorporate these considerations and cite the mentioned studies to provide a more comprehensive view of the sex bias in iRBD literature. In the Discussion, lines 748-778 now read: “However, it is important to keep in mind that in addition to differences in pathophysiological mechanisms of iRBD, other factors have been put forward to explain sex imbalance. First, the presence of a referral bias and differences in screening methods across studies have been reported.³² Second, sociocultural and environmental influences such as sleep environment and co-sleeping dynamics can influence whether symptoms are observed and reported, potentially leading to differential detection between sexes.^{56,57} Furthermore, emotional factors such as stress and anxiety seem to impact dream-enacting behaviors differently in females and males with iRBD.⁵⁸ Finally, sex differences in polysomnography measures of sleep architecture (depending on hormonal status) may contribute to differences in how iRBD manifests and is captured across sexes.⁵⁹”

References:

- 32-Li, X. *et al.* Sex differences in rapid eye movement sleep behavior disorder: A systematic review and meta-analysis. *Sleep Med. Rev.* **71**, 101810 (2023).
- 56-Kocevska, D. *et al.* Sleep characteristics across the lifespan in 1.1 million people from the Netherlands, United Kingdom and United States: a systematic review and meta-analysis. *Nat. Hum. Behav.* **5**, 113–122 (2021).
- 57-Nowakowski, S., Meers, J. & Heimbach, E. Sleep and Women’s Health. *Sleep Med. Res.* **4**, 1–22 (2013).

58-Jun, J.-S. *et al.* Emotional and Environmental Factors Aggravating Dream Enactment Behaviors in Patients with Isolated REM Sleep Behavior Disorder. *Nat. Sci. Sleep* **Volume 14**, 1713–1720 (2022).

59-Lok, R., Qian, J. & Chellappa, S. L. Sex differences in sleep, circadian rhythms, and metabolism: Implications for precision medicine. *Sleep Med. Rev.* **75**, 101926 (2024).

4. The authors made a precise introduction regarding sex-differences in neurodegeneration. However, it will benefit from a justification of the hypothesis regarding the overexpressing genes involved in estrogen-related molecular functions and the suitability of using imaging transcriptomics to investigate spatial distribution of gene expression in the brain to underlies sex-related atrophy differences.

We thank the reviewer for this comment. We agree that a more thorough description of imaging transcriptomics may strengthen the rationale for our study. The goal of imaging transcriptomics is to link the spatial distribution of regional gene expression to neuroimaging-derived brain atrophy maps (Arnatkeviciute *et al.*, 2022). Previous work, including from our group, has demonstrated the high specificity of this technique to identify genetic and molecular vulnerability factors overexpressed in brain regions showing neurodegeneration in several neurodegenerative diseases. We demonstrated that MRI-derived atrophy in Parkinson’s disease and iRBD is found in regions with high expression of genes related to mitochondrial functions and macroautophagy (Tremblay *et al.*, 2020, Rahayel *et al.*, 2023), two processes strongly affected in these disorders. Iron accumulation in Parkinson’s disease, measured through quantitative susceptibility mapping (MRI) and known to be related to the pathophysiological process of the disease, has been associated with regional gene expression of metal detoxification and synaptic function (Thomas *et al.*, 2021). In multiple system atrophy, atrophied regions on MRI showed overexpression of genes related to oligodendrocytes (Chougar *et al.*, 2025), the key cells responsible for the characteristic glial cytoplasmic inclusions associated with this disorder. We further demonstrated that Alzheimer’s disease-related brain atrophy occurred in regions enriched for protein remodelling processes, with *APOE* (gene strongly associated with Alzheimer’s disease) ranking first (Rahayel *et al.*, 2023). Recently, it was shown that the brain regions most vulnerable to declining kidney function, as assessed through structural MRI, overexpressed angiotensinogen-related genes, further highlighting the specificity and biological relevance of this approach (Rahayel *et al.*, 2024).

We now strengthen our description of imaging transcriptomic and cite these studies. The mentioned information was changed on lines 87-110, which now reads as: “To identify potential selective vulnerability factors underlying neurodegenerative changes in specific brain regions, imaging transcriptomics has emerged as a powerful approach, including in various neurodegenerative conditions.¹³ This method integrates regional neuroimaging measures with spatial gene expression profiles from post-mortem human brain atlases to uncover transcriptomic signatures associated with regional brain changes.¹⁴ In iRBD and PD, MRI-derived atrophy occurs in regions overexpressing genes linked to mitochondrial functions and macroautophagy, two processes strongly affected in these disorders.^{15,16} In PD, regional accumulation of iron measured through quantitative susceptibility mapping was related to higher regional expression of genes associated with metal detoxification and synaptic function.¹⁷ In MSA, regions with atrophy on MRI showed genetic overexpression of oligodendrocytes, the cells that harbor the glial cytoplasmic inclusions characteristic of this disorder.¹⁸ In Alzheimer’s disease, regional atrophy occurred in regions enriched for genes associated with protein remodelling processes, with *APOE*, a gene strongly associated with Alzheimer’s disease, ranking first.¹⁶ Furthermore, a recent study showed that the brain regions most vulnerable in terms of MRI atrophy to declining kidney function overexpressed angiotensinogen-related

genes,¹⁹ providing an empirical linkage for the brain-kidney axis. Taken together, this supports the ability of imaging transcriptomics to pinpoint with specificity the vulnerability mechanisms overexpressed in regions undergoing pathological changes in several neurodegenerative diseases.”

References:

13. Arnatkeviciute, A., Fulcher, B. D., Bellgrove, M. A. & Fornito, A. Imaging Transcriptomics of Brain Disorders. *Biol. Psychiatry Glob. Open Sci.* **2**, 319–331 (2022).
14. Arnatkeviciute, A., Markello, R. D., Fulcher, B. D., Mistic, B. & Fornito, A. Toward Best Practices for Imaging Transcriptomics of the Human Brain. *Biol. Psychiatry* **93**, 391–404 (2023).
15. Tremblay, C. *et al.* Brain atrophy progression in Parkinson’s disease is shaped by connectivity and local vulnerability. *Brain Commun.* **3**, fcab269 (2021).
16. Rahayel, S. *et al.* Mitochondrial function-associated genes underlie cortical atrophy in prodromal synucleinopathies. *Brain* **146**, 3301–3318 (2023).
17. Thomas, G. E. C. *et al.* Regional brain iron and gene expression provide insights into neurodegeneration in Parkinson’s disease. *Brain* **144**, 1787–1798 (2021).
18. Chougar, L. *et al.* Atrophy in multiple system atrophy relates to mitochondrial and oligodendrocytic processes. Preprint at <https://doi.org/10.1101/2025.01.22.25320961> (2025).
19. Rahayel, S. *et al.* Lower estimated glomerular filtration rate relates to cognitive impairment and brain alterations. *Alzheimers Dement. Diagn. Assess. Dis. Monit.* **16**, e70044 (2024).

5. The authors should state at the end of the introduction both the number of subjects included in the dataset (888) as the final number of subjects included in the analyses (687).

We agree with the reviewer. The mentioned information was changed on lines 274-278 which now reads as: “In this study, we used a large international, multicentre dataset of 888 brain MRI scans (408 polysomnography-confirmed iRBD patients and 480 healthy controls) to investigate sex-related differences in atrophy. Using vertex-based cortical surface analysis, we investigated the presence of a sex interaction on cortical thickness between iRBD patients and controls, on a total of 687 participants passing eligibility and quality control criteria.”

6. Methods section: The authors report that “all patients were part of ongoing prospective studies”. If the sample partially overlaps with that of previously published studies, these should be cited.

We thank the reviewer for the positive comment. The mentioned information was added and references adjusted on line 328-329, which now reads as: “Participants included in this study were part of previous multicentric work investigating brain neurodegeneration in iRBD.^{8,12,16}”

References:

- 8- Rahayel, Shady, Christina Tremblay, Andrew Vo, Ying Qiu Zheng, et al. Brain Atrophy in Prodromal Synucleinopathy Is Shaped by Structural Connectivity and Gene Expression. *Brain*. 2022; 145:9: 3162–78.
- 12- Joza, Stephen, et al. Distinct Brain Atrophy Progression Subtypes Underlie Phenotypic Conversion in Isolated REM Sleep Behaviour Disorder. *EBioMedicine*. 2025;117:105753.
- 16- Rahayel, Shady, Christina Tremblay, Andrew Vo, Bratislav Mistic, et al. Mitochondrial Function-Associated Genes Underlie Cortical Atrophy in Prodromal Synucleinopathies. *Brain*. 2023;146:8: 3301-18.

7. In the statistical analyses section of the manuscript, it is mentioned that t-tests were used. Was normality of the variables tested? T-tests assume normal distributions and non-parametric tests must be used if normality is not met. This is particularly relevant for expected non-normality in variables such as the UPDRS-III.

We thank the reviewer for this important comment. We verified the normality of variables and tested the variables with the corresponding tests. Specifically, MoCA and MDS-UPDRS-III were non-normally-distributed and retested for group differences using Mann-Whitney U tests. There were no significant differences between iRBD males and females on the MoCA ($P = 0.18$) or MDS-UPDRS-III ($P = 0.17$), as previously found. In controls, there was no significant difference between males and females on the MDS-UPDRS-III ($P = 0.93$) but unlike what was previously reported, males had lower MoCA scores compared to females ($P = 0.001$). This difference in controls does not interfere with the interpretation of the manuscript results.

We have modified the manuscript and Table 1 with the corresponding information. The Statistical Analysis section now reads as (lines 490-492): “Group differences in continuous demographic and clinical variables were assessed using independent sample t-tests for normally distributed variables, and with non-parametric Mann-Whitney U test for non-normally distributed variables. Categorical variables were compared using chi-squared tests.” Results now reads as (lines 536-540): “In terms of clinical variables, iRBD patients had higher MDS-UPDRS-III scores ($P < 0.001$) and lower MoCA scores ($P < 0.001$) compared to control participants (Table 1). Within the iRBD group, no significant differences between females and males were observed for age ($P = 0.42$), MDS-UPDRS-III scores (5.5 ± 5.2 in females vs. 6.7 ± 6.0 in males, $P = 0.17$) or MoCA scores (25.6 ± 3.9 in females vs. 25.5 ± 3.0 in males, $P = 0.18$, range: 13-30) (Table 1).”, and as (lines 549-552): “Within the control group, there were no significant differences between females and males in age ($P=0.16$) and MDS-UPDRS-III ($P=0.93$), but males had significantly lower MoCA scores compared to females (27.4 ± 2.5 in females vs. 26.8 ± 2.2 in males, $P = 0.001$, range: 19-30) (Table 1).”

8. Cluster-level significance was determined using Monte Carlo spatial permutations, with a statistical threshold of $P < 0.05$ and vertex-level significance set at $P < 0.05$, the cluster-forming threshold of 1.3 is quite liberal and increase type I error. Cluster-forming threshold of 2.0 or higher would be more appropriate.

We thank the reviewer for this comment. We chose this threshold to maximize sensitivity in detecting sex-by-group interaction effects in the context of unbalanced group sizes. Importantly, the cluster-forming step was used only to identify whether a sex-by-group interaction existed. Further imaging transcriptomic analyses used parcellated data, relying not only on significant regions, but on the complete brain map, allowing a more representative appreciation of the sex-effect characterization of atrophy.

As suggested by the reviewer, to ensure robustness, we repeated the analyses using a stringer cluster-forming threshold of 2.0. This still revealed a significant sex-by group interaction in terms of cortical thickness in the left superior parietal cortex. Analyses also still revealed two significant clusters in the left inferior parietal and postcentral cortices when directly comparing iRBD males and females that were significant at a threshold of 2.0. These clusters were also found when using the most conservative thresholds of 2.3, 3.0, and 3.3, confirming their robustness (Figure S3). We have added this information in the Results (page 17, lines 586-592), which now reads as : “When using more stringent cluster-forming thresholds to identify significant clusters ($P < 0.01$), we still identified a sex-by-group interaction effect in participants, with iRBD males showing thinner cortical thickness compared to iRBD females (Figure S3).” We also added a supplementary figure (Figure S3) to further demonstrate to the readership the robustness of our results.

Figure S3. Vertex-wise analyses of cortical thickness.
(a) Cortical map showing clusters with a significant sex-by-group interaction on cortical thickness at a cluster-forming threshold of $P < 0.01$. The colour bar indicates the statistical significance on a logarithmic scale of P -values ($-\log_{10}$), with positive values (red-yellow scale) showing a difference in iRBD males and females compared to male and female controls. (b-e) Cortical maps showing clusters with a significant sex effect on cortical thickness in iRBD at cluster-forming thresholds of $P < 0.01$, $P < 0.005$, $P < 0.001$, and $P < 0.0005$. The colour bar indicates the statistical significance on a logarithmic scale of P -values ($-\log_{10}$), with positive values (red-yellow scale) showing significant decreases in iRBD males compared to iRBD females. iRBD = isolated REM sleep behavior disorder.

9. Results: Of the 888 participants, 201 (23%) did not pass the quality control, resulting in 687 participants for analysis, please include a flow chart to specify the exclusion criteria. Additionally, the characteristics of the participants that were removed should also be reported, which was the proportion of male and female in the excluded sample?

We thank the reviewer for this comment. We have now added a flowchart to the manuscript as Figure 1. Of the 888 eligible participants (408 iRBD patients and 480 controls), 7 (0.8%) did not pass processing (4 iRBD patients, 3 controls) and 134 (15.1%) did not pass quality control (61 iRBD patients, 73 controls) based on published criteria (Monero-Sanchez et al., 2021; Klapwijk et al., 2021). Of the 65 iRBD patients, 11 were females and 54 were males. The excluded iRBD patients did not differ significantly in age ($P = 0.36$) and sex proportion ($P = 0.91$) from those passing these processing steps. The iRBD group passing processing steps significantly differed in age compared to controls (iRBD patients: 67.0 ± 6.9 ; controls: 63.7 ± 9.6 ; $P < 0.0001$). Groups were matched for age, which led to the exclusion of 60 controls (<54 years old, 25 females and 35 males), resulting in an age-matched final sample of 343 iRBD and 344 controls for analysis (iRBD patients: 67.0 ± 6.9 ; controls: 66.6 ± 6.9 , $P = 0.43$).

Figure 1. Participant selection flowchart.

The initial sample included 888 participants. Following exclusions during MRI processing, visual quality control, and age matching, the final sample consisted of 687 participants. P-values indicate differences in age distributions between groups before and after matching.

We have added this information in the Results section (page 14, lines 515-524), which now reads: “Of the 888 eligible participants (408 iRBD patients and 480 controls), 7 (0.8%) did not pass processing (4 iRBD patients, 3 controls) and 134 (15.1%) did not pass quality control (61 iRBD patients, 73 controls) based on published criteria (Figure 1 for a flowchart).⁴⁰ Of the 65 iRBD patients, 11 were females and 54 were males. The excluded iRBD patients did not differ significantly in age ($P = 0.36$) and sex proportion ($P = 0.91$) from those who passed these processing steps. However, the resulting iRBD group significantly differed in age compared to controls (iRBD patients: 67.0 ± 6.9 ; controls: 63.7 ± 9.6 ; $P < 0.0001$). Groups were therefore

matched for age, which led to the exclusion of 60 controls (<54 years old), namely 25 females and 35 males, resulting in an age-matched final sample of 343 iRBD and 344 controls for analysis (iRBD patients: 67.0 ± 6.9 ; controls: 66.6 ± 6.9 , $P = 0.43$).”

References:

40-Monereo-Sánchez, Jennifer, et al. Quality Control Strategies for Brain MRI Segmentation and Parcellation: Practical Approaches and Recommendations - Insights from the Maastricht Study. *NeuroImage*, 2021;237:118174.

10. The authors reported the sex-difference between the iRBD groups despite similar age and clinical severity. Nevertheless, we believe that the presented clinical data does not allow to support this statement and that a more extensive clinical description of the iRBD sample is needed to ensure that clinical severity is comparable: current treatments, time since iRBD diagnosis (since PSG assessment) or time since beginning of symptomatology (since self-reported initiation of iRBD symptoms) should be included. Furthermore, any olfactory assessment available for the present sample? Please, report even if it is only available for a subgroup of the study sample. Sex differences in the available sample of olfactory data should be tested to rule out the potential influence of olfactory alterations in the reported sex differences in cortical thickness. If no olfactory assessment was performed, this should be stated as a limitation and authors should preclude statements assuring comparability of clinical symptoms between groups.

We thank the reviewer for this very important comment. To address this comment, we went back to every center and gathered most of the clinical data asked by the reviewer, namely age of onset of RBD symptoms, self-reported duration of RBD symptoms at MRI, age at RBD diagnosis (PSG assessment), RBD duration since diagnosis at MRI, and information on current RBD medication. For current RBD medication, participants were considered as taking RBD medication if they were taking melatonin or clonazepam at MRI scanning. When comparing iRBD groups, we found no significant differences between iRBD females and males in any of the aforementioned clinical variables.

For olfactory assessment, every cohort performed a different test, which either was the 12-item Sniffin’ Sticks (Hummel et al., 1997), the 16-item Sniffin’ Sticks (Fedorova et al., 2020), the 40-item University of Pennsylvania Smell Identification Test (UPSIT-40; Doty et al., 1984), or the reduced 12-item version of the latter (UPSIT-12; Doty et al., 1994). First, scores were compared between iRBD males and females within centers. Next, we followed the calibration procedure developed by Lawton and colleagues to convert the different olfactory scales into a 16-item Sniffin’ Sticks score and then compare the calibrated score between iRBD females and males (Lawton et al., 2016). This showed that iRBD females had a higher score (9.4 ± 3.7 , score on 16) compared to iRBD males (8.5 ± 3.5 , score on 16), but this difference was only borderline significant ($P = 0.085$).

Added variables	iRBD females	iRBD males	P-value
Age of onset of RBD symptoms, years	59.2 ± 6.7	59 ± 9.5	0.45 ^a
Self-reported duration of RBD symptoms at MRI, years	6.9 ± 4.6	8.1 ± 7.4	0.34 ^a

Age at RBD diagnosis (PSG), years	64.8 ± 6.7	65.6 ± 7.2	0.46 ^a
RBD duration since diagnosis at MRI, years	1.5 ± 1.5	1.7 ± 2.3	0.50 ^a
Current RBD medication, % yes/% no ^b	39% / 31%	40% / 37%	0.45 ^c
Olfactory performance			
- Converted scores into Sniffin' Sticks, /16 ^d	9.4 ± 3.7	8.5 ± 3.5	0.085 ^a
- Sniffin' Sticks, 12 items (4.6% dataset) ^e	9 ± 2.6	6.6 ± 3.0	0.12 ^a
- Sniffin' Sticks, 16 items (32.5% dataset) ^e	9 ± 3.7	7.5 ± 3.1	0.09 ^a
- UPSIT-40 (35.3% dataset) ^e	23.3 ± 6.2	21.7 ± 7.2	0.25 ^a
- UPSIT-12 (27.6% dataset) ^e	7.2 ± 2.8	7.2 ± 2.9	0.5 ^a

^a Independent two-sample t-test.

^b Percentage of patients taking RBD medication (clonazepam or melatonin) vs no medication.

^c Chi-squared test comparing sex proportions of iRBD patients taking medication.

^d The 12-item Sniffin' Sticks, the UPSIT-40, and the UPSIT-12 scores were converted into a 16-item Sniffin' Sticks score following a previously developed calibration method (Lawton et al., 2016).

^e Percentage of the total dataset with available data that underwent this test.

We have now added these clinical variables in Table 1. We have modified the Methods and Results section to further mention the clinical variables collected from each centre. The Methods section (pages 7-8, lines 310-324) now reads as: “All patients underwent standardized clinical evaluations, including motor assessments using the Movement Disorders Society-sponsored Unified Parkinson’s Disease Rating Scale (MDS-UPDRS-III), global cognitive evaluation using the Montreal Cognitive Assessment (MoCA) and assessment of olfactory identification performance. Each cohort underwent either the 12-item Sniffin’ Sticks, the 16-item Sniffin’ Sticks, the 40-item University of Pennsylvania Smell Identification Test (UPSIT-40), or the reduced 12-item version of the UPSIT (UPSIT-12). For allowing comparisons between iRBD males and females, the 12-item Sniffin’ Sticks, UPSIT-40, and UPSIT-12 scores were converted into a 16-item Sniffin’ Sticks score following a previously developed calibration method.³⁹”

The Results section (page 15, lines 536-545) now reads: “No differences were also found for age of onset of RBD symptoms ($P = 0.45$), self-reported duration of RBD symptoms ($P = 0.34$), age at video-polysomnography-confirmed diagnosis of RBD ($P = 0.46$), RBD duration since diagnosis ($P = 0.50$), and proportion of patients taking RBD medication (clonazepam or melatonin) ($P = 0.45$). In terms of performance on olfactory identification tasks, there were no significant differences between iRBD males and females on each assessment scale and on the scores converted into the 16-item Sniffin’ Sticks,³⁹ although iRBD females (9.4 ± 3.7 , score on 16) tended to have better performance than iRBD males (8.5 ± 3.5 , $P = 0.085$). However, iRBD females had less years of education compared to iRBD males (12.7 ± 3.6 in females vs. 14.5 ± 3.5 in males, $P = 0.002$).”

References:

- Goetz CG, Tilley BC, et al. Movement Disorder Society-sponsored revision of the Unified Parkinson's Disease Rating Scale (MDS-UPDRS): Scale presentation and clinimetric testing results. *Movement Disorders*. 2008;23(15):2129–70.
- Nasreddine ZS, Phillips NA, et al. The Montreal Cognitive Assessment, MoCA: A Brief Screening Tool For Mild Cognitive Impairment. *J American Geriatrics Society*. 2005;53(4):695–9.
- Hummel T, Sekinger B, Wolf SR, Pauli E, Kobal G. 'Sniffin' Sticks': Olfactory Performance Assessed by the Combined Testing of Odour Identification, Odor Discrimination and Olfactory Threshold. *Chem Senses*. 1997; 22(1):39–52.
- Fedorova TD, Knudsen K, et al. Screening-Based Method for Identifying Patients with REM Sleep Behaviour Disorder in a Danish Community Setting. *JPD*. 2020;10(3):1249–53.
- Doty RL, Shaman P, Dann M. Development of the University of Pennsylvania Smell Identification Test: A standardized microencapsulated test of olfactory function. *Physiology & Behavior*. 1984;32(3):489–502.
- Doty RL, Marcus A, William Lee W. Development of the 12-Item Cross-Cultural Smell Identification Test (CC-SIT). *The Laryngoscope*. 1996;106(3):353–6.
- 39-Lawton M, Hu MTM, et al. Equating scores of the University of Pennsylvania Smell Identification Test and Sniffin' Sticks test in patients with Parkinson's disease. *Parkinsonism & Related Disorders*. 2016;33:96–101.

11. In the line of our prior comment, inclusion of variables highly relevant in terms of brain resilience as education and premorbid IQ should be reported to guarantee the groups are comparable and the results interpretable.

We thank the reviewer for this comment. We were also able to obtain information on education for the iRBD patients. iRBD males had more years of education (14.5 ± 3.5 years) compared to iRBD females (12.7 ± 3.6 years) ($P = 0.002$). Given that higher education underlies greater brain reserve and potentially brain resilience, we would have expected that iRBD males would show less neurodegeneration compared to females, which was the opposite in our study. This further aligns with our hypothesis that iRBD females appear to have selective vulnerability factors compared to iRBD males. Premorbid IQ scores were not available in any of the centers; we have added this in the limitations.

This has been added to the Results section, which now includes: “However, iRBD females had less years of education compared to iRBD males (12.7 ± 3.6 in females vs. 14.5 ± 3.5 in males, $P = 0.002$).” (page 15, lines 544-545) and in the Discussion section as an additional argument for the direction of effect (page 23, lines 793-796), which now reads as “Interestingly, iRBD females exhibit significantly less years of education compared to iRBD males in this study. Given that higher education underlies greater brain reserve and potentially brain resilience, this provides further support for the presence of selective vulnerability factors being at play in the brain of iRBD females in comparison to males.⁶²”, and as a limitation for premorbid IQ (page 25, lines 895-899), reading as “First, while the multicentric dataset is a major strength, providing a large and diverse sample, the availability of detailed clinical and demographic data, including sex versus gender distinctions, as well as premorbid IQ and hormonal status of females included in the study, was limited.”

Reference:

- 62-Valenzuela, M. J. & Sachdev, P. Brain reserve and dementia: a systematic review. *Psychol. Med.* **36**, 441–454 (2006).

12. The iRBD sample did not differ significantly regarding the mean MOCA values. However, means and standard deviations in MoCA scores in the iRBD sample suggest the presence of individuals with MCI. Providing minimum and maximum MoCA scores for each study group would contribute to a better sample characterization, as well as the MoCA cut-off that would be used to determine MCI. It would be interesting to know the percentage of MCI participants in the male and female groups, both in the iRBD and the HC. Regarding this point, the between group differences in MOCA in HC group showed a trend of significance showing lower global cognition in male participants could the results have influenced by a low-level control group for males or a higher level in the case of female-group?

We thank the reviewer for this interesting comment. We have added MoCA range for both iRBD patients and controls, as well as the number of participants in each group with MoCA score ≤ 25 . This threshold was reported as the optimal cut-off score to discriminate patients with MCI and at risk of developing DLB in iRBD (Cogné et al., 2024). We also report the percentage of MCI based on sex in each group. There was no significant difference in the proportions of possible mild cognitive impairment in males and females in either the iRBD ($P = 0.20$) or the control group ($P = 0.15$).

This information has been added to Table 1 and to the manuscript (page 15, lines 548-561), which now reads as: “Using a MoCA threshold of ≤ 25 to define possible mild cognitive impairment,⁵² the proportion of possible mild cognitive impairment was significantly higher in the iRBD group (42%; 33% of iRBD females and 44% of iRBD males) compared to the control group (22%; 17% females, 26% males) ($P < 0.001$). There was no significant sex proportion difference in either the iRBD or the control groups (iRBD: $P = 0.20$; controls: $P = 0.15$).”

Reference:

52-Cogné É, Postuma RB, et al. Montreal Cognitive Assessment and the Clock Drawing Test to Identify MCI and Predict Dementia in Isolated REM Sleep Behavior Disorder. *Neurology*. 2024;102(4):e208020.

13. To avoid false positive results, the authors should only report significant results after FDR correction, therefore the results sections should be modified accordingly including the text and figures. Moreover, the Effect size should be included in each analysis.

We thank the reviewer for this comment. However, we respectfully would like to propose a more nuanced perspective. We agree that reporting only statistically significant results after FDR correction is essential for inferential claims. However, the primary purpose of the one-sample t-tests in our study was descriptive rather than purely inferential. Specifically, these analyses were designed to illustrate the spatial distribution of deviation within each group. Here, the t-statistic reflects the magnitude of deviation from 0 (no atrophy). This allowed to provide a brain map of atrophy within each group and provided the basis for subsequent region-wise imaging transcriptomics, where all downstream analyses were strictly FDR-corrected and inferential in nature. In this sense, the t-value obtained from the one-sample test against zero can be viewed as a standardized effect size.

14. In the result section, the authors state “However, the spatial pattern of atrophy is largely similar between sex, suggesting the presence of sex-specific protective mechanisms that may mitigate brain neurodegeneration in females”. This is an interpretation of findings and should not therefore belong to Results; move to discussion.

We have moved the sentence to the Discussion section (lines 714-719), which now reads: “Our findings demonstrate that females with iRBD exhibit significantly less cortical thinning than males, despite similar age and clinical severity, with this effect absent in healthy controls. Although the spatial pattern of atrophy was largely similar between iRBD females and males, females exhibited more restricted and less severe atrophy, suggesting the presence of sex-specific protective mechanisms that may mitigate brain neurodegeneration in females.”

15. Discussion: A relevant finding of the study regards the fact that the most strongly associated gene terms were enriched in olfactory receptor activity, although this finding is not further discussed. Given the fact that anosmia is a recognized non-motor prodromal symptom in alpha-synucleinopathies and prevalent in iRBD, this finding warrants further discussion.

We thank the reviewer for their positive comment. In addition to the enrichment of estrogen-related genes, our analysis did reveal that genes associated with olfactory receptor activity were overexpressed in regions showing less atrophy in females with iRBD compared to males. Anosmia is a well-known prodromal feature of synucleinopathies and is highly prevalent in iRBD (Hu et al., 2020; Miglis et al., 2021). Sex differences in olfactory function are well established, where females generally outperform males in odor detection, identification, discrimination, and memory (Doty et al., 2009). These differences persist with age, as shown by both behavioral and neurophysiological studies (Brumm et al., 2023). Electrophysiological responses to odorants are typically stronger in females, and functional imaging studies have shown sex-specific activation patterns in olfactory-related brain regions. Furthermore, neuroanatomical data indicate that the olfactory system is sexually dimorphic, potentially driven by sex hormones, such as estrogens, which may enhance olfactory processing through increased neuronal sensitivity and modulation of synaptic transmission (Doty et al., 2009). In our cohort, iRBD females tended to perform better than males on olfactory tests, although not statistically significant. This may be due to the smaller number of females and the variability introduced by patients having been assessed on four different testing scales. Nonetheless, we agree with the reviewer that this is a worthy discussion to add to the paper, as it may be reflective of another form of selective protection in olfactory-associated regions.

We have added this to the Discussion (pages 25-26, lines 855-894), which now reads as “Our analyses also revealed that genes associated with olfactory receptor activity were overexpressed in brain regions showing less atrophy in females with iRBD compared to males. Hyposmia is a well-established feature of synucleinopathies and is highly found in iRBD.^{1,6} Notably, sex differences in olfactory function are well documented, with females typically outperforming males in odor detection, identification, discrimination, and memory.⁷² These differences persist with aging,⁷³ and have been demonstrated by both behavioral and neurophysiological studies, with stronger electrophysiological responses and distinct activation patterns in olfactory brain regions among females.^{74,75} Neuroanatomical studies also point to a sexually dimorphic olfactory system, likely influenced by sex hormones such as estrogens, which may enhance olfactory processing through increased neuronal sensitivity and synaptic modulation.⁷⁶ In this study, iRBD females tended to perform better on olfactory tests than males, although this difference did not reach statistical significance. This may be due to smaller female sample size and variability across the four olfactory testing scales used. Nevertheless, the observed transcriptomic and anatomical patterns may reflect a form of sex-specific protection in olfactory-associated regions, warranting further investigation.”

References:

- 1- Hu, M. T. REM sleep behavior disorder (RBD). *Neurobiol. Dis.* **143**, 104996 (2020).
- 6- Miglis, M. G. *et al.* Biomarkers of conversion to α -synucleinopathy in isolated rapid-eye-movement sleep behaviour disorder. *Lancet Neurol.* **20**, 671–684 (2021).
- 72-Doty, R. L. & Cameron, E. L. Sex differences and reproductive hormone influences on human odor perception. *Physiol. Behav.* **97**, 213–228 (2009).
- 73-Brumm, M. C. *et al.* Updated Percentiles for the University of Pennsylvania Smell Identification Test in Adults 50 Years of Age and Older. *Neurology* **100**, (2023).
- 74-Yousem, D. M. *et al.* Gender effects on odor-stimulated functional magnetic resonance imaging. *Brain Res.* **818**, 480–487 (1999).
- 75-Martinez, B. *et al.* Different patterns of age-related central olfactory decline in men and women as quantified by olfactory fMRI. *Oncotarget* **8**, 79212–79222 (2017).
- 76-Sanchez-Andrade, G. & Kendrick, K. M. The main olfactory system and social learning in mammals. *Behav. Brain Res.* **200**, 323–335 (2009).

16. In the discussion of their results, the authors refer to resilience in the case of females and associate the higher resilience in the group of females with estrogen gene expression in the regions of interest. Caution should be taken since the reported analyses do not control for highly relevant variables in terms of brain resilience, such as education and premorbid IQ. Additionally, this may actually differ between sexes.

We thank the reviewer for raising this important point. We agree that factors such as education and premorbid IQ are relevant contributors to brain vulnerability and could potentially differ between sexes. In our sample, as a reply to a previous comment, we found that iRBD females had less education than males (see #11), which aligns with our hypothesis. We took the decision to remove the word “resilience” throughout the paper and preferred other terms to avoid any confusion.

17. The limitations should include the lack of information regarding the hormonal status of the studied sample and its relationship with age.

We thank the reviewer for raising this. The mentioned information has been added to the limitation section of the Discussion (page 25, lines 895-899), which now reads as this: “First, while the multicentric dataset is a major strength, providing a large and diverse sample, the availability of detailed clinical and demographic data, including sex versus gender distinctions, as well as premorbid IQ and hormonal status of females included in the study, was limited.”

18. Implications of having used expression gene atlases mirroring left-hemisphere data. There is evidence suggesting lateralization of neurotransmission systems that are relevant for iRBD and related conditions (i.e. noradrenaline)

We thank the reviewer for this comment. We followed the standard approach of mirroring right hemisphere samples to their left hemispheric counterparts to maximize spatial coverage, as done in prior imaging transcriptomics studies (Romeo-Garcia *et al.*, 2018). Although some neurotransmitter systems show functional lateralization, this does not necessarily reflect gene expression asymmetries. Transcriptomic data represent potential for protein synthesis, and hemispheric differences in gene expression are generally minimal or region-specific (Karlebach & Francks, 2015). Moreover, the Allen Human Brain Atlas itself shows strong bilateral correlations in gene expression, with ~84% of genes expressed symmetrically across hemispheres (Hawrylycz *et al.*, 2012).

References:

- Romero-Garcia, R. *et al.* Structural covariance networks are coupled to expression of genes enriched in supragranular layers of the human cortex. *NeuroImage*. 2018; 171: 256–267.

- Karlebach, G. & Francks, C. Lateralization of gene expression in human language cortex. *Cortex*. 2015; 67:30–36.
- Hawrylycz, M. J. et al. An anatomically comprehensive atlas of the adult human brain transcriptome. 2012. *Nature* 489: 391–399.

19. Minor:

- a) **“and resistance to neurodegeneration resistance” (page 22). Remove one of “resistance”.**
- b) **Figure 2. Correct typo “the range of t-values for each regions”.**
- c) **only 4 regions (1%) remained significant after FDR correction, with t-scores ranging from -3.7 to -4.8 (Figure 2A). □ t-scores are shown in 2B, no?**
- d) **These findings reveal that cortical atrophy in iRBD is markedly less widespread and less severe in females compared to males ($\chi^2 = 210.65$, $P < 0.0001$), despite comparable age and clinical severity. Correct and write “is”.**
- e) **Statistics that are reported in tables should not be also reported in the text.**
- f) **PGC and other acronyms should be fully written the first time they’re mentioned**

We thank the reviewer for these comments. Changes have been made.

- g) **Table S1. Were differences corrected for multiple comparisons? This should be done and stated in each table.**

We thank the reviewer for this comment. Results shown in Table S1 were obtained and considered significant when corrected for multiple comparisons using a Monte Carlo simulation with a cluster-wise p-value threshold of $P < 0.05$. The mentioned information has been added under both Supplementary Tables 2 and 3 accordingly.

REVIEWER #3

We thank the reviewer for the positive comment.

REVIEWER #4

1. The topic is important. The work is original and the methodology is sound. The results are really interesting and the work supports the conclusions. There are some limitations, as reported by the same authors. I only suggest shortening the Discussion section and avoiding some redundancies. However, the part concerning "The findings provide mechanistic insight of sex-specific neuroprotection in prodromal stages of synucleinopathies, potentially leading to novel targeted therapeutic approaches" could be extended.

We thank the reviewer for the positive comments. We now further discuss the translational potential of our work. We have therefore added the following to the Discussion section (page 27, lines 921-932): “These findings provide new insights into sex-based neuroprotection in prodromal synucleinopathies, highlighting promising directions for targeted therapeutic strategies, as well as the importance of sex in this line of research. Indeed, the pathways overexpressed in regions less affected in females may be amenable to pharmacological modulation through selective agonists and pathway enhancers, and any neuroprotective program built on these pathways must incorporate sex as a biological variable. That iRBD females accumulate less cortical atrophy than clinically-matched males provide argument against indiscriminate pooling of sexes in trials. Stratifying randomization may yield groups with more homogenous baseline burden and progression rates, thereby increasing statistical power and reducing sample size requirements. Finally, because normative cortical thickness distributions and atrophy trajectories differ by sex, quantitative MRI endpoints used to evaluate treatment efficacy should be scored against sex-specific reference curves. Taken together, sex is an important factor impacting neurodegeneration in iRBD patients.”

REVIEWERS' COMMENTS

Reviewer #1

I thank the authors for addressing all of my previous comments and questions.

We thank the reviewer for the positive comment.

Reviewer #2

1. Thank you to the authors for addressing the majority of my previous comments. There are, however, a few minor aspects that should still be included in the revised manuscript.

We thank the reviewer for the positive comment.

2. In line with my previous comments (11 and 16), the authors should explicitly mention as a limitation the lack of information regarding the educational level of the control group. Additionally, they should include in the Discussion their rationale for not controlling for education in the RBD between-group analyses—despite having the data to do so. Specifically, they should clarify that they expected iRBD males to show less neurodegeneration than females based on the observation that RBD males had higher education levels than females.

We thank the reviewer for raising this. We want to clarify that we did not expect iRBD males to show more or less neurodegeneration relative to education level, as these descriptive analyses were added after hypothesis generation. Importantly, the reason why education was not controlled for in our analyses is because this information was not available for all sites. Specifically, some sites quantified education based on number of years, whereas others used pre-specified levels or dichotomized in >12 years or less. This is in addition to the fact that such information was not available for all controls. We have therefore adjusted the discussion, which now reads as : “Given that higher education associates with greater brain reserve and potentially brain resilience, this provides further support for the presence of selective vulnerability factors being at play in the brain of iRBD females in comparison to males.⁵⁰ In this work, education was not controlled for since the number of education years was not available for all sites. Future studies should investigate more thoroughly the impact of cognitive reserve on brain structure in iRBD” (lines 455-460) and as “This study has some limitations. First, while the multicentric dataset is a major strength, providing a large and diverse sample, the availability of detailed clinical and demographic data, including sex versus gender distinctions, as well as premorbid IQ, education, and hormonal status of females included in the study, was limited” (lines 533-536).

3. Furthermore, to enhance clarity, the authors should specify in both the Methods and Results sections the p-value threshold used to determine significance in the subcortical volumetric analyses.

P-values were added to both Methods and Results sections. The section now reads as: “For subcortical volume analyses, a statistical threshold of $P_{FDR} < 0.05$ was used to determine significance.” (lines 749-751) and “We also investigated the presence of a sex-by-group interaction on subcortical volumes and found no significant effect ($P_{FDR} > 0.05$) (Table S4)” (lines 285-287).

Reviewer #3:

See response for reviewer #2.

Reviewer #4:
The manuscript has been improved.
I have no further comments.

We thank the reviewer for the positive comment.